# Exploring Natural Deep Eutectic Solvents (NADES) for Enhanced Essential Oil Extraction: Current Insights and Applications

**DOI:** 10.3390/molecules30020284

**Published:** 2025-01-13

**Authors:** Luis Acosta-Vega, Alejandro Cifuentes, Elena Ibáñez, Paula Galeano Garcia

**Affiliations:** 1Grupo de Investigación en Productos Naturales Amazónicos (GIPRONAZ), Facultad de Ciencias Básicas, Universidad de la Amazonia, Florencia 180001, Colombia; lh.acosta@udla.edu.co; 2Laboratory of Foodomics, Institute of Food Science Research, CIAL, CSIC, Nicolás Cabrera 9, 28049 Madrid, Spain; a.cifuentes@csic.es

**Keywords:** NADES, essential oils, choline chloride, volatile compounds, GC-MS

## Abstract

Essential oils (EOs) are highly valued in the cosmetic and food industries for their diverse properties. However, traditional extraction methods often result in low yields, inconsistent compositions, lengthy extraction times, and the use of potentially harmful solvents. Natural deep eutectic solvents (NADES) have emerged as promising alternatives, offering advantages such as higher efficiency, cost-effectiveness, biodegradability, and tunable properties. This review explores the application of NADES in enhancing EO extraction, focusing on current methodologies, key insights, and practical applications. It examines the factors that influence EO extraction with NADES, including the optimization of their physicochemical properties, extraction techniques, operational conditions, and the role of sample pretreatment in improving efficiency. Additionally, this review covers the chemical characterization and biological activities of EOs extracted using NADES. By providing a comprehensive overview, it highlights the potential of NADES to improve EO extraction and suggests directions for future research in this field.

## 1. Introduction

Essential oils (EOs) are not just oils but concentrated mixtures of volatile aromatic compounds obtained from various plant parts (leaves, roots, flowers, peels, branches, bark, wood, and seeds, among others) [1,2]. These complex mixtures have diverse applications in industries such as cosmetics, medicine, and food production [3,4,5,6]. The versatility of EOs is evident in their use in perfumes, detergents, soaps, soft drinks, and pesticides, owing to their high flavoring capacity and strong fragrant effects [7,8,9]. However, interest in the biological activities of EOs has been increasing, particularly in their insecticidal and fungicidal properties. These activities are part of the broad spectrum of secondary metabolites produced by plants, which not only mitigate the effect of pathogens but also play a significant role in interactions with other plants, pollinators, and abiotic stress [10,11,12,13,14].

Furthermore, the rich and diverse composition of EOs has led to the evaluation of various biological activities for human use [15,16]. Among them, the most relevant are antimicrobial, antiviral, antifungal, and insecticidal, due to their ability to interact with and interrupt different processes related to the life cycles of pathogens [17,18,19]. The scientific community has also made significant strides in understanding the antioxidant activity of EOs and the scientific basis for their use in traditional medicine [20]. This has led to the evaluation of EOs in areas such as comfort sensation, stress management, sleep improvement, and anxiety and pain management, where they are used therapeutically in aromatherapy [21,22,23].

Various extraction approaches have been developed, given the wide range of EO applications and natural sources (Figure 1) [24]. The techniques and coadjuvants employed during the extraction process significantly influence the characteristics, components, and quantity of the essential oil obtained [25]. Although the traditional methods of steam distillation, dry distillation, or mechanical pressing are well known, new techniques have emerged [26]. For example, the use of solvents for extraction is a relatively modern approach. In this method, the plant material is mixed with a solvent and heated to a moderate temperature. After a specific time, the mixture is filtered, and the filtrate is concentrated through solvent evaporation. Finally, the EO is isolated by adding alcohol and distilling at low temperatures [27]. Solvents in this process can act as disruptive agents for the breakdown of plant cells and as preservatives for fragile volatile compounds that could degrade under the conditions of hydro-distillation or steam distillation, making them a valuable tool in the extraction of EOs [28,29]. Understanding these extraction methods is crucial for elucidating the quality attributes and physicochemical characteristics of the final product.

Although the extraction of EOs using solvents varies according to the method employed [24], there is also a need for a separation process to remove the solvent while the presence of trace amounts of the solvent in the final product limits its use, mainly if the solvents employed are toxic and/or harmful to humans, such as acetone, hexane, methanol, petroleum ether, among others [30,31]. For this reason, and to extract the highest amount of compounds while avoiding the loss of those thermolabile volatiles, other green approaches have been developed, such as the replacement of the commonly employed solvents with greener alternatives such as natural deep eutectic solvents (NADES) [32].

NADES can be defined as a natural-derived deep eutectic solvent (DES). DES consists of a mixture of two or more organic components whose melting points are significantly lower than each individual component. When both components are primary or secondary metabolites from living organisms, mainly from plants, such as sugars, amino acids, organic acids, fatty acids, or chlorine derivatives, these solvents are called natural deep eutectic solvents (NADES) [33,34,35] due to the natural occurrence of these metabolites in plants’ internal environments [36,37,38]. Since their postulation as alternative solvents by Choi et al. in 2011, NADES have gained special interest for offering a sustainable, effective, biodegradable, and practical tailoring option in the extraction and solubilization of a plethora of bioactive compounds [37,38]. This has been observed in the rapid increase in the scientific production since their postulation as extractive solvents, highlighting their properties and the advantages that they offer over conventional solvents, which continues to this day [39,40].

For this reason, the enhanced solubility and extraction capacity can be explained by the presence of the molecules that, in addition to accomplishing their primary role in the cellular environment, provide storage stabilization and solubility to secondary metabolites by mimicking the internal cell environment, which can benefit their interaction and even extraction with compounds of interest, such as EO components [41,42]. It is important to note that in addition to the improvement in the extraction of bioactive compounds, NADES are almost entirely biodegradable, sustainable, and renewable solvents due to their bio-based properties and their cheap and practical synthesis [43,44], extending and enhancing this way their use in the extraction of compounds of interest found in different plant parts that could be applied in the food, medicine, and cosmetic industries [45,46,47].

Given the crucial role of EOs and the limitations of traditional solvents in their extraction, this review explores the use of NADES as alternative solvents for extracting EOs from natural matrices. It delves into the components of NADES, the extraction techniques employed, and the factors that influence the process. Additionally, the study covers the analytical methods used to analyze EOs and their biological activities. This review aims to demonstrate NADES as a sustainable and effective alternative for extracting EOs, highlighting their potential as analytical tools and their advantages over traditional solvents.

## 2. Essential Oils: Chemistry and Extraction

The composition of essential oils is influenced by various factors, including the plant part used, environmental conditions (such as biotic and abiotic stresses), and the extraction method and its parameters [5,48,49,50]. EOs are characterized by their highly variable compositions, with each oil containing a diverse range of unique chemicals [16,51]. These chemicals contribute to the characteristic aroma and scent of the oils, which can determine their potential applications in food and cosmetics [52,53]. EOs can be classified based on various criteria, including the number of carbon atoms, biosynthetic pathways, the presence of heteroatoms (such as nitrogen, oxygen, or sulfur), and their parent backbone [51]. The primary constituents of EOs are terpenoids, which are composed of isoprene units (C_5_) and often contain oxygen. The smallest group, isoprenoids, consists of a single isoprene unit, whereas monoterpenes (C_10_) and sesquiterpenes (C_15_) are more prevalent, with sesquiterpenes being the main compounds found in EOs [54]. Diterpenes, triterpenes, and tetraterpenes are also present but typically in trace amounts. Phenylpropanoids, such as eugenol, safrole, and anethole, derived from the shikimic acid biosynthetic pathway, are another significant group of compounds found in EOs [55] (see Figure 2).

In addition to the classification of EO components based on isoprene units, these components can also be categorized into two groups: hydrocarbon terpenes and oxygenated terpenes (terpenoids). Hydrocarbon terpenes are composed exclusively of carbon and hydrogen atoms, while oxygenated terpenes contain oxygen in various functional groups, such as alcohols, ethers, aldehydes, oxides, phenols, and esters [18,51]. Furthermore, the structures of terpenoids vary and include acyclic, monocyclic, bicyclic, tricyclic, and even aromatic compounds. Terpenoids are highly nonpolar, insoluble in water, and soluble in pure alcohols [25,56]. In addition to their structural diversity, terpenoids are known for their promising biological activity. They have demonstrated potential in therapeutic studies owing to their anticancer, antioxidant, antibacterial, antiviral, and immune-boosting properties [57,58,59,60].

**Figure 2 molecules-30-00284-f002:**
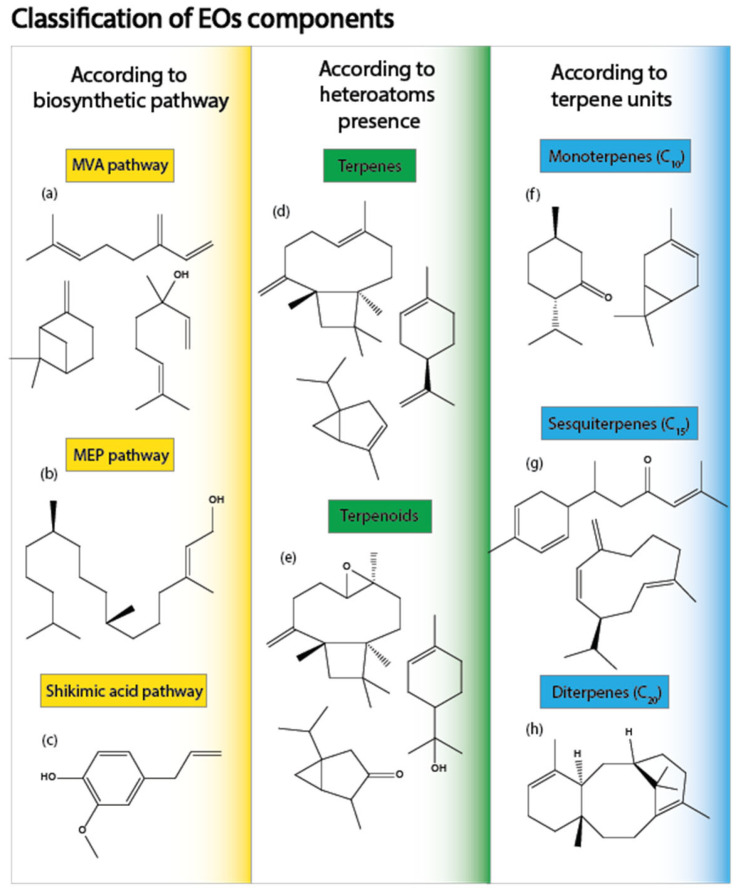
Different classifications of EOs components. MVA: mevalonic acid; MEP: 2-C-methylerythritol-4-phosphate, also called the non-mevalonate pathway. Components of EOs: (**a**) β-myrcene, β-pinene, and linalool, components of *Citrus arantium* leaves EO [61]; (**b**) phytol precursor of carotenoids and phytosterols in plants [61,62]; (**c**) eugenol, one of the main component of *Syzygium aromaticum* buds and *Cinnamomum cassia* bark EOs [63], (**d**) caryophyllene, limonene, and α-thujene, terpenes composed exclusively of carbon and hydrogen [16]; (**e**) caryophyllene oxide, terpineol, and thujone, oxygenated analogous of terpenes [56]; (**f**) menthone and 3-carene, examples of a monoterpenoid and a monoterpene, respectively [64]; (**g**) α-turmerone and germacrene D, examples of a sesquiterpenoid and a sesquiterpene, respectively [65]; (**h**) taxadiene, a precursor diterpene of the anticancer drug paclitaxel [66].

Essential oils are extracted using various techniques, typically categorized into conventional and advanced methods. Traditional extraction methods include steam distillation (SD), hydro-distillation (HD), hydro-diffusion (HDF), and solvent extraction (SE) [67]. On the other hand, advanced methods comprise microwave-assisted extraction (MAE), ultrasound-assisted extraction (UAE), and supercritical fluid extraction (SFE) [24]. The increase in the use of advanced methods for EO extraction is due to the limitations of conventional methods, which include low efficiency, long extraction times, the use of (toxic) organic solvents, and high energy consumption [68,69]. Although the advanced methods overcome many of these disadvantages, greener approaches have been developed for EO extraction. These include methods such as microwave-assisted hydro-distillation (MAHD) and ultrasound-assisted hydro-distillation (UAHD), which combine the high-energy transfer of advanced techniques with traditional hydro-distillation. This reduces extraction time, carbon footprint, the use of non-food or non-cosmetic grade solvents, and the energy demands of conventional methods (Figure 1) [70,71,72].

## 3. Application of NADES in EOs Extraction

The use of NADES to extract bioactive compounds from plants is closely linked to the composition of these mixtures, due to their mimetic behavior in the internal cellular environment [73,74]. According to the literature, many bio-based organic compounds are used as hydrogen bond donors (HBDs) and hydrogen bond acceptors (HBAs) in the formulation of NADES (Table 1, Figure 3). The hydrogen bonds formed between the components result in a liquid at room temperature, which is characterized by low volatility, low toxicity, and high solubility of phytochemicals [75,76]. Additionally, one of the most remarkable aspects of NADES is their high tunability, which can be tailored for different applications such as extraction, synthesis, and analysis [44]. This tunability modifies their chemical properties (such as polarity), enabling interactions with specific compounds like the components of EOs through non-covalent interactions, such as hydrogen bonding and electrostatic (repulsive and attractive) interactions [77,78]. In the analyzed literature, different NADES compositions have been studied for the extraction of EOs and even the extraction of some EO components that represent particular interest, such as the terpene lactones from *Ginkgo biloba* L. leaves EO [79], phenylpropanoids like myristicin, safrole, and methyl eugenol from *Myristica fragrans* Houtt. EO [80], and eugenol from clove (*Syzygium aromaticum* L.) buds EO [81]. Regarding the constituents of NADES, the most diverse composition was found in the HBDs, with 29 different organic compounds used to formulate NADES (Figure 3). In contrast, there was less variation in the composition of HBAs used for EO extraction, with only 16 different compounds, and choline chloride (ChCl) as the most commonly used HBA in these formulations (Figure 3).

The formation of intermolecular hydrogen bonds between NADES components creates a network that facilitates the solubilization of molecules with different polarities [76,82]. Although the primary interaction is due to hydrogen bonding, the presence of charge delocalization from the basic/acid functional groups in the different components allows other intermolecular forces, such as electrostatic and dipole–dipole forces, to contribute to the extraction of compounds with varying polarities [73]. Additionally, it is important to highlight that, besides the green chemistry advantages of NADES, their ability to modulate factors such as viscosity, pH, conductivity, and polarity makes them a promising alternative for the extraction of bioactive compounds [83]. Regarding polarity, changes in the NADES components, their molar ratios, and water content significantly influence the composition and efficiency of extraction [41,84,85].

For the extraction of terpenoids from *Coriandrum sativum* L. (seeds), a series of ChCl-based NADES were evaluated. Glucose, glycerol, urea, and citric acid were used as HBDs. Additionally, the HBA/HBD ratio was evaluated (1:1, 1:2, 1:3, 2:1, 3:1) to screen the most suitable solvent for extracting the highest amount of EO (1.10% when using NADES compared with 0.77% using water as an extracting solvent under optimal conditions) and also the highest concentration of linalool, α-terpineol, and limonene [85]. Similarly, Tang et al. used ChCl-based NADES formulations to extract linalool, α-terpineol, and terpinyl acetate from *Chamaecyparis obtusa* leaves using ethylene glycol (EG) as the HBD in different ratios and analyzed via headspace solvent microextraction (HS-SME) [86]. It was observed that the extraction of the three compounds increased with the ChCl/EG ratio, with ChCl/EG 1:4 extracting the highest amount of the target molecules. Both studies serve as examples of the versatility and the advantages that NADES offer in the extraction of target compounds. In both cases, ChCl-based NADES are used for extraction. Due to the moderate-to-high polar nature of the solvents, compounds with similar polarity will be extracted, such as terpenoids, which were observed in the extraction of linalool in the highest proportion, followed by α-terpineol in both studies. These compounds possess a hydroxide group that will interact with the hydroxide group present in ChCl as well as those present in the HBDs used, such as glucose, ethylene glycol, and glycerol. Therefore, an increase in the proportion of hydrophilic compounds extracted is expected, as was observed in the proportion of oxygenated hydrocarbon (OHC) terpenes such as linalool when compared with hydrocarbon (HC) terpenoids like limonene. On the other side, when the target compound lacks oxidized moieties, such as the presence of oxide, alcohol, aldehyde, or even organic acid functionalities, and it is composed exclusively of carbon and hydrogen, the NADES component can be tailored in order to prioritize the extraction of such compounds [87,88,89]. For example, in the extraction of β-caryophyllene from New Zealand Manuka leaves (*Leptospermum scoparium*), twenty-six types of hydrophilic and hydrophobic NADES were evaluated, and the extraction with conventional solvents (hexane, ethanol, methanol, and water) was evaluated [84]. Initially, there was a notorious difference between the hydrophilic and the hydrophobic NADES in the extraction of β-caryophyllene, the latter being the ones with a higher extraction amount and even similar to conventional solvents such as ethanol. This is attributed, according to the authors, to the polarity of the NADES, since β-caryophyllene is lipophilic and therefore will dissolve in non- or low-polar solvents, showing a low extraction in ChCl-based NADES and a great extraction in quaternary ammonium salts and menthol-based NADES, confirming that changes in the HBA switch the polarity medium and consequently greatly influence the extraction of compounds of interest. It is also important to mention that although the HBA modifies the compounds that will be extracted in great proportion, HBD affects the extracting performance through changes in the viscosity, as was observed when extracting in the extraction yield when 1-dodecanol (yield = 2.16 mg/g Manuka leaf) was replaced with tetradodecanol (1.17 mg/g Manuka leaf) in a tetrabutylammonium chloride-based NADES. The authors suggest that this is due to an increase in the viscosity, which limits the mass transfer and results in a minor amount extracted. Although the organic salt-based NADES showed extracting yields similar to conventional solvents such as hexane and methanol, these were still lower than the menthol-based NADES, which showed the greatest extraction yield (even higher than the two previous solvents mentioned), which is attributed to its superior hydrophobic and organic characteristics. Additionally, it is important to mention that the size and molecular weight of the HBD also impact the extraction performance, which also explains the higher extraction obtained with lactic acid when compared with other HBDs such as levulinic acid, lauric acid, and capric acid [84]. A higher molecular weight and size may increase the viscosity and therefore limit the interaction with the target compounds, decreasing the overall extraction [90,91]. The previously mentioned proves the flexibility of NADES for being tailored to extract target compounds from plant matrices such as terpenes and terpenoids, tailoring physicochemical parameters such as the viscosity and polarity through the chemical composition of the solvent components, and even designing specific approaches according to the research objective since these compounds possess important biological properties and a potential industrial application [92,93,94].

Enhancing the composition and yield of EOs is one of the main approaches to using NADES for solvent extraction. In extracting EO from *Angelica sinensis* radix, Fan et al. evaluated six chlorine chloride-based NADES at the same molar ratio, and for comparison purposes, they carried out the same experiments using water as the extracting solvent [32]. The results showed that the weight of the EO and the process yield were higher when NADES was used as the extraction solvent (1.07%) compared with water as the extracting solvent (0.75%) under MHD conditions. In addition, the contents of *Z*-ligustilide and *E*-ligustilide were higher in the EO obtained using NADES, which represents an advantage of this methodology, given the high instability and pharmacological importance of these compounds.

One of the main advantages of using NADES as a solvent in the EO extraction is the vast and diverse components that can be employed as HBA, HBD, or both. In the extraction of EO from perilla leaves (*Perilla folium*), choline chloride, fructose, and glucose were used as HBA, while glucose and fructose, as well as glycerol, malic acid, and citric acid, were used as HBD in mixtures with different molar ratios (1:1, 1:2, 1:3, 2:1, 3:1) [95]. The EO yield obtained using water was the lowest among all the solvents evaluated (0.17%), in comparison to the highest yield obtained using ChCl/MA (0.76%) at a molar ratio of 2:1. Furthermore, there is higher chemical variability in the composition of perilla EOs obtained with NADES than that obtained with water. Additionally, although the three principal components of the EOs were the same (perillaldehyde, β-caryophyllene, and α-bergamotene), their relative abundance varied with each NADES group: Perillaldehyde was higher in pure water (PW), followed by Fru/MA and ChCl/Gly. β-caryophyllene was extracted in a more significant proportion in ChCl/gly and ChCl:Fru, while the presence of α-bergamotene was higher in ChCl/Gly, followed by Ch:Glu and ChCl/MA. Finally, some compounds, such as α-terpineol and damascenone, were exclusively found in EOs extracted with NADES.

Because of the high versatility of different compounds that can act as HBA/HBD in NADES composition and the tunability of these solvents through the modulation of their molar concentrations, optimization of the solvent is one of the common approaches related to the extraction of EOs. In the extraction of EO from leaves of *Mentha haplocalyx* Briq. carried out by Li et al. [96], authors screened the most suitable solvent by switching among ten different HBDs and evaluated an enzymatic hydrolysis pretreatment of the plant material to obtain the highest extraction yield and an increase in the content of target compounds such as menthol, menthone, piperitone, and iso-menthone. The results showed that NADES not only contributed to solubilizing the components of EOs but also the enzymes present in the solution, allowing the solvent to reach the plant matrix through the disruption of the cell wall combined with the energy absorption capacity, resulting in a higher release of the target compounds [96]. This was observed in the extraction yield where the enzyme-deep eutectic solvent pretreatment followed by microwave-assisted hydro-distillation showed the greatest value (2.56%) followed by and deep eutectic solvent microwave-assisted hydro-distillation (2.19%), then enzyme-pretreatment followed by microwave-assisted hydro-distillation with water as extracting solvent (1.84%), followed by conventional hydro-distillation (1.69%) and lastly, microwave-assisted hydro-distillation (1.60%). Although the use of ChCl as HBA is widely extended, other HBAs have also been studied for the extraction of EOs, such as the work carried out by Mori et al. on the extraction of EO components from tea tree (*Melaleuca alternifolia*) and lemon grass leaves (*Cymbopogon citratus*) [97]. In this study, betaine-based DES were used and combined with polyols and organic acids such as glycerol, lactic acid, sucrose, and citric acid. The results showed that a higher yield in the extraction of EO components of tea tree, such as eucalyptol and terpinolene (0.049 and 0.099%, respectively), was observed when using NADES (composed of betaine and sucrose 2:1) compared to when using conventional solvents such as water (0.026 and 0.0086%, respectively) and ethanol (0.025 and 0.040%, respectively). Additionally, the same NADES formulation was also useful in the extraction of neral, geranial, and geraniol from lemongrass (0.084, 0.27, and 0.006%) and showed higher values compared with ethanol (0.043, 0.16, and 0.005%).

Although EOs contain a wide variety of chemical compounds, some extractive approaches target specific compounds of particular interest that are present in the sample. Therefore, the isolation and enrichment of EOs with these substances represent an objective that can be fulfilled using NADES. Deterpenation is usually one of the processes required for the purification of citrus essential oil (CEO) [52], which is intended to reduce the presence of limonene in this EO. Li et al. carried out a deterpenation process based on the in situ formation of NADES in essential oils. For this, quaternary ammonium salts were used as HBA to extract linalool, a highly valuable terpenoid in CEO, and to separate it from limonene indirectly. Results showed that the associative extraction approach reached a linalool yield extraction of 89.25% [98]. A similar approach was used to extract different compounds employing NADES as solvents, such as β-caryophyllene from New Zealand Manuka (*Leptospermum scoparium*) leaves [84], terpene lactones (ginkgolides A, B, C, J, K, and bilobalide) from *Ginkgo biloba* L. leaves, and even the isolation of a new epoxysesquiterpene from *Ageratina adenophora* flowers [99] (Table 1).

A common pattern observed in the tailoring of NADES is the variability in their composition, according to the compounds that are expected to be extracted. This explains the use of compounds such as proline, menthol, lauric acid, capric acid, 1-dodecanol, eugenol, and L-carnitine (Figure 3), switching the polarity of the chemical environment and allowing low-polarity compounds to be dissolved and released from the plant matrix [100]. Although, as already mentioned, NADES components play an essential role in the extraction of EOs and influence their composition, other critical aspects of EO extraction must be addressed to make the extraction process efficient, economical, and ecologically sustainable. The following section discusses the factors that influence EO recovery, chemical composition, and extraction performance.

**Table 1 molecules-30-00284-t001:** Applications of NADES in the extraction of EOs and their components.

Plant Specie/Part Employed	Selected NADES and Extraction Technique and Conditions		Results/Main Outcomes	Main Compounds Identified	Analytical Technique	Reference
*Flos Chrysanthemi Indici*/Flowers	Bet/Gly (1:2) at 70 °C with stirring liquid–solid ratio at 21.5 mL/g. Introduce CO_2_ into the liquid at a flow rate of 0.25 L/min	High-speed homogenization coupled with an SHS approach combined with CO_2_ and NADES extraction Rotating speed at 20,712 rpm, and extraction time of 3.23 min.	EO yield: 1.23%. High values in the extraction of TPC, TFC, and EO. Higher content of OHC	Cycloeucalenol acetate Hedycariol Cryptomeridiol	GC-MS	[101]
*Schisandra chinensis (Turcz.) Baill*/Fruits	ChCl/EG molar ratio 1:3 and solvent ratio 7:3 *v*/*v*, melted in a 80 °C oil bath. S/L: 1:30 *w*/*v*. 10.0 g of dried fruit were mixed with 300 mL, 0.75 M of DES.	Alternate UMHD at 20 min of extraction time. 400 W (UE), 250 W (MW)	EO yield: 12.2 mL/Kg. Higher free radical scavenging activity and reducing power and OHC proportion in EO composition	Ylangene α-bergamotene β-himachalene	GC-MS	[102]
*Angelica sinensis*/Roots	ChCl/CA (1:3). Water content of 40%. S/L: 1:5 *w*/*v*.	MAHD at 99 °C and the reaction time at 70 m. 600 W (MW).	EO yield: 1.39%. Higher EO yield, efficiency, and low energy consumption compared with traditional and water-based methods	*Z* and *E*-ligustilide, n-butylphthalide, 3-*n*-butylidenephthalide, (+)-Isovalencenol	GC-MS	[32]
*Leptospermum scoparium*/Leaves	Hydrophobic deep eutectic solvent (HDES) Me/LA (1:2). 0.75 g of Manuka powder (200 m) was mixed with 5 mL of HDES	Stirring. The mixture was stirred at 1000 rpm for 1 h at 25 °C	EO yield: n.r. Higher antioxidant and antibacterial activity as well as high TPC compared with traditional solvents	β-caryophyllene α and β-pinene, linalool, eucalyptol, α and β-selinene, α-eudesmol, α-terpineol	GC-MS	[103]
*Cymbopogon citratus* *Cymbopogon flexuosus* *Elyonurus muticus*/Leaves	ChCl/AA (1:2)	Magnetic stirring method at 60 °C, 180 min for the species *C. citratus* and *C. flexuosus* and 150 min for *E. muticus* species.	EO yield: n.r. Higher antioxidant, antibacterial activity, and high TPC in the NADES-based EOs compared with traditional solvents (H_2_O, MetOH, EtOH, and Ace)	Neral Geraniol Geranial *p*-menthol *p* and *o*-cymene Citronellol	GC-MS/MS	[104]
*Coriandrum sativum* L./Seed	ChCl/Ur (1:1) At 80 °C 75 g of NADES with 40% (*w*/*w*) water in a 1:5 (*w*/*w*) plant/liquid ratio.	US pretreatment followed by conventional HD. (US) at T = 25 °C, t = 30 min, and 70 W of power. Hydro-distillation for 4 h.	EO yield: 1.10% NADES positively influenced EO composition and extraction yield. The acidity of the NADES used may influence the composition of the CEO. Operational extraction conditions were optimized successfully through RSM	Linalool Limonene α-terpineol	GC-MS GC-FID (for chiral analysis)	[85]
*Perillae folium*/Leaf	ChCl/MA (2:1). 72 g of the components mixed with 48 g deionized water were placed in a beaker with an 80 °C water bath and 100 W US for 15 min L/S: 40:1	US pretreatment followed by conventional HD. Ultrasonic extraction instrument (600 W, 5 min). In the next step, the mixture was diluted with 800 mL deionized water and put under US at 600 W for 20 min.	EO yield: 0.67% NADES components ratio influenced EO yield, TPC, antioxidant, and antibacterial activity	α-terpineol Damascenone Perillaldehyde β-caryophyllene α-bergamotene	GC-MS	[95]
*Litsea cubeba* (Lour.) Pers./Fruits	ChCl/OA (1:1) Water content (50%), liquid–solid ratio (12.5:1 mL/g)	NADES-homogenate-based MAHD. Homogenate time (2 min), and microwave power (700 W).	EO yield: 16.49% NADES led exclusively to extracting specific compounds, and the homogenate pretreatment with NADES enhanced EO yield. Antioxidant activity in NADES-based EOs was higher than those without pretreatment and water-based experiments	*m*-cymene *trans*-linalool oxide *Z* and *E*-citral Eucalyptol	GC-MS	[105]
*Piper nigrum*/Fruits	ChCl/Fr (3:2). Nades was prepared with small amount of water at 80 °C. L/S: 3:1 DES/powder ratio	Three-stage extraction: MW pretreatment, fast heating stage, and MAHD. (1) pretreatment stage: 600 W microwave power, 80 °C temperature, and 10 min duration; (2) fast heating stage: 600 W microwave power, 110 °C temperature, and 5 min duration; (3) hydro-distillation stage: 300 W microwave power, 110 C temperature, and 35 min duration.	EO yield: 1.78% The different stages and their parameters influence EO composition and yield. The optimized extraction approach allowed the identification of a higher number of compounds than HD and MAHD (solely)	Caryophyllene Eucalyptol Sabinene α-pinene	GC-MS	[106]
*Aloysia* *Citriodora*/Leaves	ChCl/Glu (1:1) Pretreatment: (100 g) and 80 g of NADESs were mixed with 240 mL of distilled water. HD: water ratio of 1:10	MW pretreatment at power (600 W) and time (5 min) followed by HD.	EO yield: 0.21% MW power during the pretreatment stage influenced EO yield, and the antioxidant and antimicrobial activity varied according to NADES components	Verbenone Limonene Spathulenol	GC-MS	[107]
*Mentha haplocalyx* Briq./Leaves	ChCl/Glu stirred at 80 °C with 80% water Content. S/L: 1/12 g/mL	Enzyme-based (cellulase and pectinase 1:1, 2.0% enzyme concentration) NADES pretreatment followed by MAHD at 540 W Pretreatment temperature: 50 °C	EO yield: 2.19% The synergistic effect of the enzyme pretreatment and NADES during extraction significantly improved the components and yield from EO. Additionally, the EO extracted under those conditions showed an α-amylase and AChE inhibitory activity superior to those extracted with traditional methods and without pretreatment	Menthol Menthone Piperitone Isomenthone Germacrene D	GC-MS	[96]
*Artemisia absinthium*/Leaves	Car/MA (1:1) with 50% water S/L: 1:6	SD Distillation time of 10 h	EO yield: 7.52% NADES inclusion in the extraction showed a higher extraction yield and a higher number of identified compounds than water-based, NaCl, and enzymatic treatments	α-terpineol l-borneol 1,8-cineole Thujone	GC-MS	[108]
*Ipomoea cairica* (L.) *Sweet*/Leaves	ChCl/Glu (1:1) stirred at 85 °C with 15% water S/L: 1:5	Heating and stirring Rotating speed: 300 rpm, rotating time: 25 min	NADES solvents have shown to be efficient in the extraction and dilution of plant components and valuable for the analysis of volatile compounds	β-elemene β-caryophyllene α-humulene	Static HS-GC-MS	[109]
*Myristica* *fragrans* Houtt./Seeds	ChCl/CA (1:1) with 40% water (*w*/*w*) stirred at room temperature and subsequently sonicated at 25° C for 10 min. S/L: 1:15	US pretreatment followed by HD. Pretreat: the suspension of nutmeg fruits and 40% NADES at 50 C for 30 min. Distillation time: 2 h	EO yield: 1.41% NADES inclusion improved EO yield compared with water, and the proportions of sesquiterpene hydrocarbons, monoterpene hydrocarbons, oxygenated monoterpenes and phenylpropanoids varied according to NADES composition	Elemicin Methyl eugenol Safrole Myristicin	GC-MS	[80]
*Amomum kravanh*, *Amomum tsaoko Amomum villosum*/Fruits	ChCl/EG (1:4) stirred at 80 °C. L/S: 7:1	Three-stage extraction: MW pretreatment, fast heating stage, and MAHD. (1) pretreatment stage: 500 W of MW, 50 °C of temperature, and 7 min of duration; (2) fast heating stage: 600 W of MW, 110 °C of temperature, and 5 min of duration; (3) MAHD stage: 300 W of MW, 110 °C of temperature, and 30 min of duration.	EO yield: 3.64, 2.16, and 1.62%, for *A. kravanh*, *A. tsaoko*, and *A. villosum* The optimized method allowed a higher number of compounds to be identified than HD and MAHD (solely) and contributed to the differentiation of the three species according to their EO components	Eucalyptol Isobornyl formate Camphor	GC-MS	[110]
*Syzygium aromaticum*/Buds	ChCl/LA (1:2) with deionized water (20% *w*/*w*) and exposed MW (400 W) at 80 °C by (10–15 min. Pretreatment stage: 30 g of clove bud powder, 80 g of DES	Three-stage extraction: MW pretreatment, fast heating stage, and MAHD. Pretreatment stage: MW (600 W), temperature (80 °C) and reaction time (5 min). Fast heating stage: MW (600 W), the temperature (110 °C) and the reaction time (5 min). MAHD stage: 300 W of MW, 110 °C of temperature, and 40 min of duration.	EO yield: 4.60% The optimized method brought more compounds than others, such as HD, MAHD (solely), and water-based MAHD. Additionally, the method employed was more effective and environment friendly according to CO_2_ emissions and electrical consumption calculations	Eugenol β-caryophyllene Eugenyl acetate α-humulene	GC-MS	[111]
*Curcuma longa* L./Roots	ChCl/OA (1:1) exposed to microwave (400 W)at 80 °C. S/L: 1:2	Three-stage extracti. on: MW pretreatment, fast heating stage, and MAHD. Pretreatment stage: MW (600 W), temperature (84 °C) and reaction time (5 min). Fast heating stage: MW (600 W), the temperature (110 °C) and the reaction time (5 min). MAHD stage: 300 W of MW, 110 °C of temperature, and 76 min of duration.	EO yield: 0.85% Method condition parameters optimized led to a higher extraction of EO and allowed the identification of a higher number of compounds than MAHD and HD methods	ar-turmerone α-turmerone α-himachalene	GC-MS	[112]
*Ageratina adenophora*/Flowers	ChCl/LA stirred at 85 °C. 20 g dried and powdered plant material and 72 g of Des with 850 mL of distilled water	US pretreatment followed by HD. US: at 25 °C for 25 min hydro-distillation: 70 °C, 2.5 h.	EO yield: 13.52% The incorporation of NADES not only enhanced EO yield but also influenced the chemical composition. EOs obtained with NADES led to the isolation of a new sesquiterpene in high yield and showed a potential AChE inhibitory activity	5,11-epoxycadin-3,4-en-8-one Bornyl acetate β-bisabolene	GC-MS ^1^H and ^13^C NMR SEM SC-XRD	[99]
*Nardostachys jatamansi* (D.Don) DC/Roots	ChCl/MA (2:1) heated at 90–100 °C with stirring. Ratio plant material, NADES, and water: 1:2:4 (*w*/*v*/*v*).	HD; 70 °C for 3–4 h	EO yield: 1.77% *v*/*w* The NADES components highly influenced the chemical composition of EOs and, consequently, biological activities, such as insecticidal activity against *A. craccivora* and *P. lilacinus* and inhibitory activity of AChE and glutathione *S*-transferase	Bisabolol α-cadinol Nootkatone Valeranone Nerolidol	GC-MS GC-FID	[113]
*Rosmarinus officinalis* L./Leaves	ChCl/Gly (1:2) with 10% of water S/L 1:15 *m*/*v*	NADES soaking followed by HD. Pretreatment at 20 °C during 72 h. 2 h of distillation.	EO yield: 2.32% The pretreatment stage enhanced the EO yield, quantitative composition, and antioxidant activity compared with the non-pretreated sample	Camphor Verbenone Borneol	GC-MS GC-FID	[114]
*Cuminum cyminum* L./Seeds	ChCl/LA (1:3) with 40% (*w*/*w*) of water L/S: 6:1	Three-stage extraction: MW pretreatment, fast heating stage, and MAHD (1) pretreatment stage: 600 W of MW, 90 °C of temperature, and 4 min of duration; (2) fast heating stage: 600 W of MW, 110 °C of temperature, and 5 min of duration; (3) MAHD stage: 300 W of MW, 110 °C of temperature, and 30 min of duration.	EO yield: 2.22% NADES inclusion resulted in more EO extraction and a greater number of compounds identified, increasing the OHC proportion in the overall composition. Additionally, MW showed to be more suitable for sample pretreatment than US	Cuminol Cuminal Moslene Terpineol	GC-MS	[115]
*Mentha piperita* L./Leaves	ChCl/Glu (5:2) diluted with water (70% *w*/*w*) S/L: 1:10	UAE (maximum power of ~500 W, ambient temperature)	The use of NADES allows the identification of volatile compounds without the use of time-consuming methods and, organic solvents and can be applied to the differentiation of peppermint samples from different origins	Menthol Menthyl acetate Pulegone Menthone Eucalyptol	HS-SPME-GC-MS	[116]
*Ginkgo biloba* L./Leaves	ChCl/Asc diluted with deionized water (34%). S/L: 1:10	UAE (temperature of 56 °C, 37 min)	NADES influenced the extraction of the different ginkgolides, with the highest total yield that one obtained by the selected NADES (24.60%) and allowed the lowest ginkgolic acid extraction (0.37%) Additionally, NADES solvents showed higher efficiency than traditional solvents such as MetOH and EtOH	Bilobalide Ginkgolide A, B, C, J, and K Ginkgolic acid	HPTLC-MS	[79]
	Bet/EG (1:3) containing 40% (*w*/*w*). S/L: 1/10	UAE at 45 °C and 100 w for 20 min	Extraction yield: 2.36% Using NADES in the extraction led to a higher yield than traditional solvents like EtOH and H_2_O, as well as traditional methods such as HD and solvent-reflux extraction	Triterpene lactones	HPLC-ELSD	[117]

n.r.: not reported; GC-MS: gas chromatography coupled with mass spectrometry detector; GC-FID: gas chromatography coupled with flame ionization detector; HPLC: high-performance liquid chromatography; HS: headspace; NMR: nuclear magnetic resonance; SC-XRD: single-crystal X-ray diffraction; SPME: solid-phase microextraction; HPTLC: high-performance thin-layer chromatography; ELSD: evaporative light scattering detector; SHS: switchable-hydrophilicity solvent; TPC: total phenolic compounds; TFC: total flavonoid compounds; EO: essential oil; OHC: oxygenated hydrocarbons; MW: microwave radiation; US: ultrasound radiation; HD: hydro-distillation; SD: steam distillation; UMHD: ultrasound/microwave-assisted hydro-distillation; MAHD: microwave-assisted hydro-distillation; UAE: ultrasound-assisted extraction; RSM: response surface methodology; AChE: acetylcholinesterase.

## 4. Factors Affecting the Extraction of Essential Oils with NADES

As described previously, NADES offers high versatility and advantages in the extraction of EOs when compared with conventional solvents; however, other parameters of the extraction process, such as the extraction method, extraction conditions, use of adjuvants, and conditioning of the sample, also play a critical role in the observed results. Therefore, to continue exploring the factors affecting EOs extraction, NADES properties, extraction techniques and conditions, and sample pretreatment will be discussed in the following subsections.

### 4.1. NADES Properties

Selecting the solvent components is part of the screening for the most suitable solvents for extracting specific compounds present in the plant material. Other parameters, such as the molar ratio and water content in NADES, are crucial since they determine the formation of the hydrogen bond network and its strength in the extraction [38,118]. Therefore, optimizing these parameters to enhance the extraction yield and enrich the chemical composition (e.g., increase the proportion of terpenoids) is one of the main objectives during the extraction of EOs [16].

In the extraction of EOs from *Angelica sinensis* radix, six different ChCl-based NADES were evaluated (ChCl/Glu, ChCl/CA, ChCl/EG, ChCl/Gly, ChCl/MA, and ChCl/Xyl) with the same molar ratio (1:2) and the same water content (30%) under the conditions of assisted microwave hydro-distillation (NADES-AMHD) [32]. The results showed that ChCl/CA had the highest EO extraction yield and weight, which can be attributed to the presence of organic acids that better absorb microwave energy, converting it into heat, which disrupts cell walls and enhances solvent penetration into the plant matrix. This was equally evident during the extraction with ChCl/MA and ChCl/Gly [32]. In the extraction of EO from *Coriandrum sativum* L. using ChCl-based NADES, a total of 20 different NADES were evaluated by changing the HBD (Glu, Gly, Ur, and CA) and their molar ratios (1:1, 1:2, 1:3, 2:1, and 3:1) [85]. The concentration of EO varied significantly among the different NADES; ChCl/Ur (1:1) presented the highest EO extraction yield (0.95%), while the molar ratio did not have any significant effect. In addition, the water content in NADES was optimized, where 30% water content resulted in the highest extraction yield (1.10%). The high viscosity of NADES is one of the main challenges when using these solvents due to the formation of hydrogen bonds, which can be assessed using infrared spectroscopy (FT-IR), where shifts at lower or higher frequencies demonstrate non-covalent interactions between the molecules [41,119]. The addition of water is a common alternative to overcome viscosity problems. However, this can negatively affect the performance of NADES because of the perturbation or alteration of the network formed between HBD and HBA. Therefore, this parameter should be optimized for NADES preparation and subsequent extraction procedures [120].

Both the yield of essential oil (EO) and its composition, including the proportions of specific compounds within the EO, are equally important for optimizing NADES properties. During the extraction of EO from *Litsea cubeba* (Lour.) Pers. dried fruits, eight ChCl-based DES were combined with different HBD (OA, Glu, TA, CA, MA, Mal, Ur, and Gly) [105]. The amount of EO extracted was highly influenced by NADES type, with oxalic acid having a positive influence on the extraction process and showing the highest extraction yield (9.48 mg/g), followed by glucose (6.73 mg/g), malic acid (6.47 mg/g), and citric acid (6.45 mg/g). 

This is explained by the different roles of HBD under microwave radiation in promoting the release of compounds present in the plant matrix, such as microwave absorption, cell wall disruption, and cellulose dissolution [121]. NADES type also influenced the composition of EO; in this sense, it was observed that although the main compounds in the EO were the same in all the tested solvents (linalool, *Z*-citral, and *E*-citral), other compounds were extracted in a higher proportion, such as borneol in TA/ChCl, CA/ChCl, and MA/ChCl; caryophyllene in Glu/ChCl; patchouli alcohol in Ur/ChCl and Gly/ChCl; and *m*-cymene and *trans*-linalool in OA/ChCl [105].

The molar ratio also influenced the yield of EO, with a higher yield when the OA/ChCl ratio was 1:1, which decreased when the molar ratio was reduced (1:4–1:2), and it was attributed to the interactions between the components of the EO and NADES. Furthermore, the chemical composition of EO was also affected by molar ratio composition, increasing the extraction of α-phellandrene, α-terpinene, *m*-cymene, eucalyptol, γ-terpinene, and trans-linalool oxide with increasing the molar ratio from 1:4 to 1:1 (OA/ChCl) [105]. Finally, the water content in NADES affected the extraction yield and chemical composition, and the amount of EO extracted increased similarly to the water content in the solvent, reaching a maximum value at 50% and strongly reducing the amount of EO at higher values, which was attributed to the reduction in the interactions between the NADES components and the target compounds. Additionally, a low water content extracted more terpenes such as α-terpinene, *m*-cymene, and γ-terpinene. At the same time, an increase in this parameter allowed for higher extraction of oxygenated terpenes such as trans-linalool oxide, piperitenone, α-terpineol, eucalyptol, and α-terpineol [105].

In the work of Hu et al., four different HBAs (other than ChCl) were evaluated (ChCl, Bet, Car, and TEAC) for the extraction of wormwood (*Artemisia absinthium*) EO [108]; eight different HBDs (Glu, EG, PG, Gly, TA, and MA) were tested. The results showed that NADES composed of L-carnitine and malic acid presented the highest extraction yield (4.53 mg/g) among all the NADES prepared and even higher than the process that used water as the solvent (3.76 mg/g) and pretreatment with enzymatic hydrolysis (3.89 mg/g). This was confirmed by SEM analysis, where a deeper damaged surface was observed after the extraction process with Car/MA compared to the other NADES and techniques, showing the capacity of the solvent to dissolve cell wall structure and promote the release of compounds inside the cell [108,122]. A similar trend was also observed in the extraction of EO from tea tree (*Melaleuca alternifolia*) and lemon grass (*Cymbopogon citratus*) using Bet-based NADES [97].

Similarly, different studies highlighted the use of NADES in the extraction of EOs and carried out different optimization approaches to screen for the most suitable NADES, varying the nature of HBAs and HBDs [79,80,84,107,112,113,116], properties such as the molar ratio of the components and water content [99,115], paying special attention to the efficiency and selectivity of the process. For example, in the extraction of ginkgolides and ginkgolic acids from *Ginkgo biloba* leaves, 15 different NADES were evaluated, varying their composition (HBA and HBD nature) and keeping the water content in the NADES and the molar ratio between the components constant [79]. Results showed that the NADES type highly influenced the extraction of different ginkgolides; for instance, ChCl/AA showed the highest extraction of total ginkgolides, especially ginkgolide A. This behavior was observed for ChCl-based NADES and was attributed to the high degree of hydrogen bonding and electrostatic interactions of the ginkgolides and the green aspect of choline chloride, highlighting and promoting its use as a NADES component. Finally, the different NADES were compared with traditional solvents (EtOH, MetOH, water, and polyethylene glycol), where the ChCl/AA NADES extracted a higher amount of ginkgolide A, C, K, J, B, and bilobalide (10.43, 3.93, 2.02, 2.54, 2.39, and 8.10 mg/g, respectively) than traditional solvents, such as water (2.82, 0.0, 1.10, 0.82, 0.71, and 4.00 mg/g, respectively), and also showed a lower amount of undesirable ginkgolic acids [79].

In the context of essential oil (EO) extraction, the use of NADES has emerged as a promising alternative to traditional solvents. However, it is important to note that many studies in this area focus on the performance of NADES as a whole solvent system, without isolating the individual components (HBAs and HBDs). This approach, while useful for assessing the overall efficacy of NADES in extracting EOs, limits the ability to critically evaluate the impact of different HBA/HBD combinations on the extraction process. Additionally, the combinations of NADES components used in EOs extraction studies are often diverse and complex, with many studies employing formulations that are not directly comparable. Various HBA/HBD pairs, including choline chloride (ChCl) in combination with sugars, acids, and other compounds, have been tested. However, the lack of standardized formulations and consistent experimental conditions makes it difficult to draw general conclusions about the relative effectiveness of different NADES systems for EO extraction. Moreover, while there is growing interest in exploring alternative HBAs to ChCl, the majority of studies in EOS extraction focus on ChCl-based NADES. Research on non-ChCl-based alternatives is still limited, and those studies that do explore these alternatives often do not provide sufficient data to enable a comprehensive comparison of their extractive performance. As a result, a critical analysis of alternative NADES formulations, particularly in the context of EO extraction, remains difficult to conduct with the current body of literature. Given these limitations, this review primarily focuses on ChCl-based NADES, as they have been more extensively investigated in EOs extraction and provide a more consistent foundation for comparison. However, it is important to acknowledge that the field would benefit from further research aimed at standardizing NADES formulations and isolating the effects of individual components. Such studies would enable more robust and meaningful comparisons across different NADES systems, thus advancing our understanding of how to optimize these solvents for enhanced EO extraction.

### 4.2. Extraction Conditions

The dissolving power and penetrating capabilities of NADES in the extraction of EOs have been demonstrated previously; however, their viscosity is one of the main challenges to overcome in the extraction of these bioactive compounds [123,124]. Additionally, water use can negatively affect the intermolecular interaction network between NADES components and target compounds, influencing the yield and chemical composition of the extracts [37,125]. An alternative that is extensively used to overcome this disadvantage is the use of energy to heat the solvent, decreasing its viscosity and increasing the contact between plant material and NADES [126]. Regarding EOs, although there is a distinction between conventional and advanced methods, NADES application usually involves using both types. Also, it is worth mentioning that the combination of both aims to optimize time, materials, and energy [127].

Eight different ChCl-based NADES were evaluated using a two-part extraction approach: pretreatment and extraction of EO from *Mentha haplocalyx* Briq leaves [96]. According to our findings, microwave power is one of the leading energy sources used to extract EOs with NADES because of advantages such as significant energy density, enhanced dilution and permeability of the plant material, and fast heating transfer [128]. However, microwave irradiation power, irradiation time, and parameters associated with extraction are also crucial in the extraction process. In EO extraction from *Mentha haplocalyx* Briq leaves, four different irradiation powers were evaluated (230, 385, 540, and 700 W), showing that the maximum microwave power was not suitable for extraction, possibly because of the potential degradation of compounds present in the EO, which explains the use of an irradiation of 540 W since it showed the highest yield (1.93%), followed by 700 W (1.50%), 385 W (1.48%), and 230 W (1.36%). Additionally, the relative concentration of the compounds in the EO increased with increasing irradiation power. The S/L ratio is another important factor in the extraction process. Generally, EO yield increased with the volume of extraction solvent to plant material ratio (S/L). In the mentioned example, five different S/L ratios were evaluated. A trend in extraction yield was observed where the increase in the S/L ratio resulted in a higher extraction yield, reaching its maximum value at 1/12 g/mL (2.20%). Similarly, the relative concentrations of the EO components (menthone, d-limonene, piperitone, menthyl acetate, and germacrene D) also followed the same behavior, reaching higher relative abundances with increasing S/L ratios [96].

In the same study, the effect of the extraction method was also evaluated by comparing the optimized enzyme-deep eutectic solvent pretreatment followed by microwave-assisted hydro-distillation (EDP-MAHD) with traditional hydro-distillation (HD), MAHD, enzyme-pretreatment followed by microwave-assisted hydro-distillation (EP-MAHD), and deep eutectic solvent microwave-assisted hydro-distillation (DES-MAHD). Results showed that the conventional HD showed a higher yield than MAHD (1.69 and 1.60%, respectively) which indicates that although the heating methods differ in the extraction yield, the use of microwave power not resulted in a positive effect net during the extraction process, however, according to the kinetic model, the kinetic constant was higher in MAHD which indicates that the energy transfer was superior than conventional hydro-distillation and although was not evaluated in the study, the efficiency of the energy transfer during the extraction was superior to the conventional HD since microwave power can increase the temperature in a lower time through heating the internal extraction medium first since needing to heat the environments and then transfer it to the extraction medium through convection which is required with conventional heating (HD) [129]. On the other hand, microwave power, NADES, and enzymatic pretreatment (EDP-MAHD) resulted in a synergistic positive effect that was observed in the efficiency of the extraction process, reaching its peak yield in less than 20 min and the highest concentration among the evaluated methods (2.56%). The hypothesis behind this behavior is that the enzyme may continue to be active after pretreatment and, therefore, in the presence of NADES, the compounds released from the plant matrix form an equilibrium with the solvent components that promote its solubilization [130]. Lastly, the chemical composition of EOs was also affected by the extraction method, varying the relative intensity of compounds such as menthol, menthone, piperitone, iso-menthone, menthyl acetate, and germacrene D.

Generally, it is necessary to optimize the conditions during the extraction process to improve and enhance the composition and yield of the extracted EO. For this purpose, the design of experiments is one of the main tools that allows the study of the relationship between independent variables (input variables) and responses (output variables). Therefore, to optimize the extraction time and energy consumption, avoid the degradation of thermal-sensitive interesting compounds, and enhance extraction yield, the use of the design of experiments is convenient and widely employed in this area of research. For example, during the extraction of terpene lactones from *Ginkgo biloba* L. leaves, ultrasound (US) was used as an energy source, and parameters such as extraction time, water percentage, and extraction temperature were the independent variables with three different levels, while EO yield and the concentration of the ginkgolides (A, B, C, K, J, and bilobalide) and ginkgolic acid were the responses in a Box–Behnken experimental design, with a total of 17 experiments with 5 replications at the center point for pure error estimation and reproducibility confirmation [79]. During the optimization, a non-linear quadratic model is generated for every response, and the linear, quadratic, and interactive effects of the independent over the response are represented by a polynomial equation. The analysis of variance (ANOVA) is used to determine the significance of the regression model’s terms towards the responses, analyzed through the F and *p* values. Additional validations, such as lack of fit and R^2^ and predicted R^2^, are used to validate if the model describes the relationship between the experimental factors and the response and if it fits the experimental data, respectively [131,132].

Statistical analysis showed that the water percentage during the extraction had a negative significant impact on ginkgolides extraction (both linear and quadratic terms, *p* < 0.0001), which is expected since the high viscosity of NADES may limit the mass transfer of the target compounds from the plant matrix to the extractive medium. The EO yield and the amount of ginkgolides extracted increased remarkably until a water percentage of 34%, reaching their highest values, and decreased afterward (until 50%), which is associated with the break of the hydrogen bonds’ network from NADES and consequently the loss of extractive ability. As mentioned previously, the viscosity of NADES is one of the main disadvantages of using these types of solvents for extraction, since it contributes to a low diffusion rate and low mass transfer, being one of the main disadvantages to overcome during extractive approaches. Ultrasonication temperature (both linear and quadratic terms, *p* < 0.0001) showed a positive effect on the extraction yield of the different ginkgolides, decreasing the viscosity and surface tension and promoting the release of compounds in the plant material into the dissolving media. In this study, the optimum temperature was 57.6 °C. Finally, ultrasonication time showed less appreciable effects on ginkgolides extraction than the first two parameters, showing a less appreciable increase in the extraction of ginkgolides when testing sonications times between 10 and 60 min. Due to this and in order to reduce operational times and save energy, an extraction time of 36.63 min was selected as the optimal condition.

Regarding the statistical terms, the lower *p*-value and higher F-value indicate the influence of the factor on the studied response, with the water content being the one with the highest value, followed by sonication temperature and finally sonication time in the linear terms. Furthermore, the quadratic term of water content was the only one that showed a significant effect on the ginkgolides extraction, while the interaction of water content with sonication temperature showed the highest, followed by the interaction of water content with time, and lastly, the crossed effect of ultrasonication time and temperature. In addition to the statistical terms for the optimization of the extractive process, there are other tools that can contribute to finding the optimum conditions, such as the perturbation plots, 3D response surfaces, contour plots, and the desirability function, which for the present study, led to finding the optimal conditions (water content of 33.656%, ultrasonication time of 36.636 min, and temperature of 55.774 °C) [79].

Similar approaches have been developed in order to optimize the extracting conditions as it was observed in the extraction of coriander EO where a central design (CD) was used to optimize the EO yield with plant to NADES ratio (X_1_), water percentage (X_2_), pretreatment temperature (X_3_) and time as factors (X_4_) [85]. The model was significant (F-value of 13.66 with *p*-value < 0.001 and no significant lack of fit, *p* = 0.6832). Additionally, there was a good correlation between the predicted EO yields and the experimental data (R^2^ = 0.97), and the parameters that showed a significant impact on the response studied were X_1_, X_2_, and X_3_ (*p* < 0.01), while the pretreatment time linear term did not show a significant value (*p* > 0.05). Similarly, the quadratic terms of pretreatment temperature and plant to NADES ratio showed an extremely significant effect on the response (*p* < 0.01), while water percentage had a significant impact (*p* < 0.05). According to software analysis, the optimal conditions for extraction are as follows: X_1_ = 1:4, X_2_ = 30%, X_3_ = 40 °C and since the pretreatment time did not reveal a significant impact on EO yield and with the aim of following green chemistry principles, a pretreatment time of 30 min was selected. The EO yield obtained under these conditions was 1.10%, which showed a significant difference from the yield obtained with the NADES-assisted UAHD (0.94%) and conventional HD (0.77%), indicating that the optimization produced a significant increase in the EO yield and, since this is very close to the predicted value, highlights the use of these models for optimizing the extraction of EOs with NADES.

During the extraction of EO from dried fruits of *Litsea cubeba* (Lour.) Pers., in addition to the optimization of the NADES solvent type, liquid/solid ratio (L/S), microwave power, and extraction time under MAHD-NADES conditions, the technique was compared with conventional HD and MAHD (water based) [105]. Results showed that high S/L values improved EO extraction, ranging from 8.0 mg/g of EO to 12.0 mg/g when the S/L increased from 5 mL/g to 12.5 mL/g. This is attributed to the amount of NADES present being sufficient to dissolve the cellulose in the cells and, therefore, release the compounds from the inside of the cell. There was a slight difference between the selected value (12.5 mL/g) and the highest value tested (15.0 mL/g), indicating that the amount of solvent present was sufficient and there was no increase in extraction yield by further increasing the L/S ratio. Microwave power eases cellulose disruption and release of EO components into the medium. A low microwave power (230 W) yielded a lower extraction than a higher microwave power (700 W), where the extraction increased by more than 50%. Similarly, EO composition varied according to the extraction conditions; for example, an initial decrease followed by a significant increase in α-phellandrene, *m*-cymene, isopulegol, α-terpineol, and caryophyllene was observed when increasing L/S values. Microwave power and time also influenced EO composition; when microwave power increased, *m*-cymene, limonene, eucalyptol, γ-terpinene, trans-linalool oxide, and caryophyllene decreased significantly first and then increased, reaching equivalent and, in some cases, higher values than by using the highest microwave power (700 W). This represents an advantage in the extraction of these compounds because they are not degraded and can be obtained with high extraction yields. Microwave time affected the extraction of α-phellandrene, *m*-cymene, D-limonene, eucalyptol, and γ-terpinene, which were not detected during an extraction time of 10 min and were extracted with prolonged time. Conversely, caryophyllene, α-terpineol, and piperitenone levels were reduced when the microwave time increased.

As previously mentioned, the operating conditions influence the yield and composition of the final product during EO extraction. In addition, the technique employed significantly affects the extraction process. It offers advantages such as protection of the compounds of interest, shorter extraction times, and easy separation/isolation of the final product [27,133]. The same study compared HD, water-based MAHD, and NADES-based MAHD. Although the EO yield using HD was higher compared with MAHD (7.41 and 4.37 mg/g, respectively), HD required a longer time than the other techniques to start boiling and separating the EO from the medium compared with MAHD (35 min and 3.38 min), highlighting the use of advanced methods such as MAHD. Also, it was observed that NADES had a good impact on the overall extraction, reaching the highest yield (16.49 mg/g) in the shortest time (3.12 min), which highlight the use of NADES during the extraction that, combined with MAHD, allows a higher energy and mass transfer, promoting the release of EO to the extracting medium from the plant material [105].

Similar approaches have been used for the extraction of EO; for instance, in the extraction of terpenoids from *Codonopsis pilosula* Franch, the S/L ratio, temperature, and time were optimized, and the techniques evaluated were MAE, UAE, and both coupled (UMAE) techniques [134]. Similarly, during the extraction of *Angelica sinensis* radix EO, the effects of NADES, method, and microwave power were assessed [32]. Results showed that NADES significantly increased the EO yield compared to HD methods (NADES-HD and water-based HD). Moreover, long HD extraction times (5 h) are highly undesirable, and microwave power reduces extraction times and increases yield. The difference between NADES-based MAHD and water-based MAHD extraction yields (1.073 and 0.751%, respectively) is attributed to the penetration capacity of NADES constituents, which can absorb higher microwave radiation than water and convert it into thermal energy, dissolving cell wall components and extracting soluble substances from the plant material [135,136,137]. The different methods evaluated also affected the chemical composition and quantity of the EO. Although the main components were present in the four evaluated methods (*Z*-ligustilide, *E*-ligustilide, *n*-butylphthalide, and 3-*n*-butylphthalide), NADES-based MAHD showed the highest yield of *Z* and *E*-ligustilide (81.68 and 8.72%, respectively) compared with the water-based extraction (80.35 and 7.81%, respectively), highlighting the advantages of using NADES in the extraction of one of the main compounds present in *Angelica sinensis* EO and with a particular interest due to its pharmacological properties [138].

Once the compounds have been extracted in the medium, it is necessary to have a separation step to isolate the EO for analysis and quantification purposes. This step is highly important since it can play an important role in the quantity of EO recovered and its chemical composition [76]. In most of the studies reviewed, hydro-distillation is the technique employed for the separation of EOs once the extraction with NADES has been performed. This could be attributed to the combination of advanced techniques such as microwave-assisted and ultrasound-assisted which extract higher efficiency compounds such as EOs components but still require the use of conventional techniques for their isolation like hydro-distillation since the chemical nature of these compounds allows them to move to the gas phase for being collected, also, the temperature-stable nature of NADES components and their extremely low volatility makes this method suitable since only terpene-like compounds will be separated and ultimately, the modern extraction apparatus allow multiple functionalities in one single device and the extraction and separation occur in different stages on the same device, promoting the use of hydro-distillation as separation technique. However, as was seen before when comparing different techniques in the extraction of EOs, HD requires long extraction times with high temperatures and the presence of water, which could be a threat for thermal-labile compounds, and therefore it must be optimized in order to minimize a decrease in EO yield as well as the compounds of interest present. For example, in the extraction of black and white pepper (*Piper nigrum*) plus the optimization of the NADES type, NADES/plant material ratio, and conditions related to the pretreatment, such as microwave power, temperature, and time, HD time was also optimized [106]. The results showed that EO yield increased with HD duration, showing a great increase in the first 25 min (1.72%), then followed by a moderate increase until 35 min (1.77%), and finally no substantial increase from this time, so this time was selected for the optimization since the purpose of the extraction is to reduce the overall time extraction. Similarly, during the extraction of wormwood EO, four different HD times were evaluated, finding that at 10 h duration of HD showed the highest EO yield (7.5 mg/g), and an extended period of time (12 h) led to a reduction in EO yield (6.5 mg/g) [108]. This, according to the authors, could be attributed to the loss of volatile compounds of EO components due to the extended period of time and also the formation of hydrosol since the large presence of water. It is important to mention that for reaching HD conditions, it is necessary to reach the water boiling point (100 °C); therefore, this hypothesis seems highly suitable to the experimental conditions for HD separation. The trend of reaching a high value and then remaining stable has also been observed in other studies where the changes in EO yield can be considered negligible, such as in the extraction of EO from *Amomum kravah,* where at 30 min of HD the EO yield was 3.48 while at 60 min of HD it increased slightly to 3.51% [110]. This behavior was also observed in the extraction of EO from cumin (*Cuminum cyminum* L.), where no noticeable improvement in EO yield was observed after 30 min of HD, and in the extraction of EO from clove buds, after 40 min, the extraction yield remained stable [111,115]. Lastly, regarding this separation technique, it is important to mention that after separation, water could remain in the final product, and therefore extra steps are necessary for its removal, such as the use of desiccants, the use of freeze-drying, or organic solvents such as hexane or ethyl acetate, which increases the cost of the overall process or the presence of other undesirable chemical agents.

Other different approaches for the separation of EO components from the extraction medium have been developed, such as the use of centrifugation and filters, like in the separation of β-caryophyllene from Manuka leaves [84]; stirring and separating with organic solvents such as ethyl ether, ethyl acetate, and hexane to induce the formation of organic and aqueous layers, the former where the EO components will migrate, as was evidenced in the extraction from hinoki (*Chamaecyparis obtusa*), clove (*Syzygium aromaticum*) buds, peppermint (*Mentha piperita*) leaves, tea tree (*Melaleuca alternifolia*), and lemon grass (*Cymbopogon citratus*) [81,97,116,139]. Another method employed for the separation of EO components from the extraction medium is the application of solid-phase extraction (SPE), where the compounds of interest are concentrated in cartridges that possess a high affinity for these kinds of compounds, which was observed in the extraction of terpene lactones from *Ginkgo biloba* L. leaves, where after ultrasound radiation was applied, the extractive medium was centrifuged and the supernatant purified with HLB cartridges [79].

Regarding the extraction parameters and conditions, the cost associated with extracting EOs and its environmental impact is not only necessary for the green aspect associated with the use of NADES for the extraction, but also it is necessary to address the energy consumed during the process and calculate its effect in the environment. Among the studies gathered in the present review, few presented an analysis regarding the energy consumption and the cost associated with extracting EOs; for example, during the extraction of EO from *Litsea cubeba* (Lour.) Pers. fruits, conventional HD required the highest amount of time (65 min) compared with MAHD (34.30 min) and NADES-MAHD (33.12 min). This was reflected in the electricity consumption of the three methods (0.76, 0.40, and 0.39 kWh for HD, MAHD, and NADES-MAHD, respectively) and related to the amount of EO extracted, which allows for estimating the cost of production per quantity of EO, with NADES-MAHD being the one with the highest yield (37.67 mg/g/kWh), followed by MAHD (18.43 mg/g/kWh) and lastly HD (8.08 mg/g/kWh). Additionally, the CO_2_ production was also affected by the extraction method, with NADES-MAHD producing the lowest value (309 g), followed by MAHD (320 g), and HD with the highest value (607 g). As can be seen, advanced techniques combined with NADES not only reduce the environmental impact but also are efficient in energy consumption, offering a higher extraction yield and a short time for the isolation of EO [105]. The same conclusion was obtained during the extraction of EO from clove buds and cumin seeds, where the electrical consumption and the carbon footprint of the NADES-based extraction with advanced techniques (such as MW and US) were lower compared with water-based extraction and conventional extraction methods (HD) [111,115]. Additionally, since NADES are almost chemically inert solvents, and their separation from extracted solutes is one of the objectives in the extraction of EOs, NADES solvents can be re-used in different cycles for extracting a higher amount of solutes [140]. Among all of the studies, only one reported the use of NADES consecutively in the extraction of β-caryophyllene, showing a significant increase from 3.32 mg/g in the first cycle to 6.07 mg/g in the second cycle and 8.77 mg/g in the third cycle and keeping steady in the fourth cycle [84]. This proves the reusability of NADES in the extraction of EO components and also highlights the use of these kinds of solvents that, in addition to their biodegradable, low-volatile, and almost non-toxic characteristics, can be used more than one time, reducing the amount of NADES necessary for extraction [37,141]. As mentioned before, these types of analyses are highly important since, in addition to the green aspect of NADES, the search for efficient and green extraction and isolation approaches should be one of the main objectives related to the extraction of EO with NADES, and then, since these kinds of analyses were scarce in the studies included in the present review, we encourage future researchers in the field to include aspects related to the environmental impact, costs, and time of EO extraction in their works.

As previously mentioned, extraction conditions play a significant role in the extraction of EO when using NADES, specifically in the yield and chemical composition of the final product. However, according to the literature retrieved, in addition to the parameter conditions and NADES properties, sample pretreatment is another aspect related to the extraction of EO with these green solvents, which is discussed in the following section.

### 4.3. Sample Pretreatment

A three-stage procedure was evaluated for extracting EO from *Cuminum cyminum* L.: a pretreatment step, followed by a fast heating step, and finally, a hydro-distillation step [115]. The extraction technique was designed to maximize the extraction of EO and reduce extraction time; the pretreatment stage promoted the release of EO components from the plant material, the fast heating step increases the temperature of the medium to reach boiling temperature, and hydro-distillation is employed to separate EO from the medium. In addition to optimizing parameters such as NADES composition, water content, and L/S ratio, three other methods were evaluated: microwave hydro-distillation (MHD), ultrasound-assisted NADES pretreatment combined with microwave hydro-distillation (UA-NADES-MHD), and microwave-assisted NADES pretreatment coupled with microwave hydro-distillation (MA-NADES-MHD). For comparative purposes, the difference between the evaluated methods resides exclusively in the pretreatment stage: MA-NADES-MHD employs NADES and microwave radiation during the pretreatment stage. At the same time, MHD uses deionized water instead of NADES and simultaneously serves as a control with the former method, whereas UA-NADES-MHD employs ultrasonic radiation during the pretreatment stage. The subsequent stages (fast-heating and hydro-distillation) had the same operating conditions [115].

The results again showed the superiority of NADES in absorbing microwave radiation, with the yield of the EO extracted with MA-NADES-MHD (1.89%) superior to the MHD method (1.13%), with deionized water as the solvent. Similarly, microwave radiation is more convenient than ultrasound technology during pretreatment, with an EO yield of 1.57% for the UA-NADES-MHD method [115]. Once the technique to be employed is selected, parameters during the pretreatment step are optimized since they can influence the final yield, composition, and isolation of EO. For example, in the present study, parameters such as time, temperature, and microwave power were optimized during the pretreatment step. The MW power enhances cellular disruption through the vibration of molecules, contributing to the dissolution and release of volatile compounds by cellulose. This was observed since increasing the microwave power led to an increase in EO yield. However, when the microwave power was set to 600 W, the EO yield reached its maximum. A decrease in EO yield was observed at higher microwave power levels, which was attributed to the degradation of volatile compounds due to excess energy [142]. Temperature enhances the interaction between NADES and microwave radiation, where the increase in temperature contributed to increased EO yield; however, the highest yield was obtained at 90 °C, and a decrease in the response was observed to a further increase in temperature, which is attributed to the evaporation of EO components. Pretreatment duration, therefore, could be defined as the reaction mixture at the highest accessible temperature in which no evaporation has occurred (e.g., the loss of volatile compounds is minimal) [115]. Here, equilibrium was found after 4 min, and the extraction yield was almost constant. This time was selected under optimum conditions to avoid degradation or reduction in the EO yield.

Once the optimal working conditions were identified through the experimental design, they were evaluated, resulting in an average essential oil (EO) yield of 2.22%. Qualitative and quantitative analyses of EO were conducted and compared with the other two techniques (MHD and UA-NADES-MHD). This study identified 87 compounds using the three methods. The MA-NADES-MHD method identified the highest number of compounds (58), followed by UA-NADES-MHD (48) and MHD (45). The main components found in the EO of cumin seeds were cuminol, cuminal, moslene, terpineol, and β-pinene, with their relative proportions varying based on the extraction method. Notably, cuminol was present in the EO but has not been reported in the literature reviewed. According to the authors, the chemical composition of EO is influenced by experimental factors, such as the raw material and the three-stage microwave extraction. The use of NADES during the extraction allowed the recovery and further identification of additional compounds, including *E*,*E*-2,6-Dimethyl-1,3,5,7-octatetraene, *p*-mentha-1,4-dien-7-ol, and α-himachalene. Unique compounds, such as copaene, 2-carene, α-santalol, tricyclene, and β-cymene, were exclusively found using the MA-NADES-MHD method. Regarding the relationship between hydrocarbons (HC) and oxygenated hydrocarbons (OHC), a higher OHC proportion is desirable for the EO composition. Although the UA-NADES-MHD method achieved the highest OHC proportion (87.10%), the EO yield was 1.57%. In contrast, the MA-NADES-MHD method, with a lower OHC proportion of 78.93%, yielded 2.22% of EO. This higher EO yield with MA-NADES-MHD resulted in a larger absolute amount of OHC, demonstrating its greater effectiveness in extracting essential oils from cumin seeds.

Another example is the extraction of EO from coriander (*Coriandrum sativum* L.), in which pretreatment time, temperature, and plant-to-NADES ratio (S/L) were optimized using response surface methodology (RSM) with a central composite design (CCD) while applying ultrasound radiation [85]. The results indicated that pretreatment temperature and S/L ratio were significant (*p* < 0.05) factors in increasing EO yield. However, no significant differences were observed in the chemical composition or enantiomeric fractions of linalool.

Similar approaches have been developed to optimize the extraction yield and enhance EO’s chemical composition (e.g., increase OHC) by optimizing pretreatment parameters, such as pretreatment time, temperature, energy source type, and power. Two non-exclusive trends were observed regarding the inclusion of the pretreatment stage during the extraction process and the use or absence of NADES during this step. First, pretreatment significantly impacts extraction yields compared to traditional methods such as hydro-distillation (HD), water-based microwave-assisted hydro-distillation (MAHD), and ultrasonic-assisted hydro-distillation (UAHD). For instance, in the extraction of black and white peppers (*Piper nigrum*), the use of ChCl/Fru (3:2) as a solvent led to substantial improvements in both yield and efficiency, reducing extraction times and costs, reaching yields of 1.78 and 1.77% for white and black peppers, respectively, higher values compared with water-based pretreatment (1.72 and 1.71%, respectively) and conventional hydro-distillation (1.69 and 1.64%, respectively), and therefore, highlighting the pretreatment stage as a positive step prior to extraction [106]. Although the overall chemical composition of the EO was more diverse with the pretreatment and NADES approaches, there were no significant changes in the proportions of the main compounds. In some cases, pretreatment with NADES led to reductions in these main compounds. This suggests that NADES may facilitate the transformation of these compounds within the medium and help prevent their evaporation. Consequently, although more EO was extracted, the OHC proportion did not vary significantly among the methods (22.75, 23.14, and 21.03 for DES-MHD, MHD, and HD, respectively, for white pepper and 21.25, 19.9, and 19.84%, respectively, for black pepper). This indicates that the transformation of compounds in the pretreated samples resulted in substances that were not detected in EOs obtained through traditional methods. A similar behavior was observed in the extraction of EO from rosemary (*Rosmarinus officinalis* L.) leaves when using *glyceline* (ChCl/Gly 1:2) as NADES and soaking for 72 h as pretreatment [114] and EO from three *Amomum* species (*A. kravanh*, *A. tsaoko*, and *A. villosum*) using ChCl/EG (1:4) as NADES and microwave power in the pretreatment stage [110].

On the other hand, pretreatment effects with NADES were positively reflected in chemical composition diversity and extraction yield. This could be attributed to two consecutive mechanisms: the interaction of NADES with the plant material and the interaction of the plant components (now released into the medium) with NADES under radiation conditions. In the first, the strong absorption capacity of organic compounds, such as NADES components, and their ability to transform it into thermal energy promote cellular disruption and, combined with their high solubilizing capacity, release a large number of compounds in the medium. This is observed, for example, with the increase in EO yield and the number of compounds identified, as was observed in the extraction of EO from clove (*Syzygium aromaticum*) buds, where ChCl/LA (1:2) as NADES was used during the microwave-assisted pretreatment [111]; EO extracted from turmeric (*Curcuma longa* L.) with ChCl/OA (1:1) as NADES combined with microwave-assisted pretreatment [112]; and during the extraction of EO from *Ageratina adenophora* flowers using ChCl/LA (1:3) and ultrasound-assisted pretreatment [99]. Increases in EO yield of 58.6, 44.0, and 376.0% were observed for *S. aromaticum*, *C. Longa* L., and *A. adenophora*, respectively, when NADES was used during the pretreatment stage. Similarly, a significant increase in the number of compounds identified was found in the three studies, and some compounds were identified exclusively in the NADES-based pretreatment, unlike the water-based analogous pretreatment.

In the second mechanism, once the EO components are in the extraction medium, they can undergo chemical transformations due to NADES components, water molecules from NADES, radiation (microwave or ultrasound), and high temperatures. This was observed not only through the detection of new compounds but also through changes in HC/OHC values, such as the increase in the percentage of oxygenated compounds in the extraction of EO from clove buds, from 46.7% in the water-based MAHD to 53.3% in the NADES-based MAHD. Compounds such as eugenol, β-caryophyllene, eugenyl acetate, and α-humulene were the main components of the EO and varied significantly between the methods. This behavior was also observed for the HC/OHC ratio and ar-turmerone and α-turmerone in turmeric EO and the epoxysesquiterpene 5,11-epoxycadin-3,4-en-8-one in *A. adenophora* flowers, whereas the latter was only found in the NADES-based EO. These qualitative and quantitative changes in EO composition could be associated with the transformation of the chemical compounds present in the plant. For example, linalool from coriander EO can undergo different modifications and rearrangements due to its chemical nature (allylic tertiary alcohol), which can be transformed into other compounds such as β-ocimenol, myrcenol, α- or β-terpineol, and limonene; the last two are highly related to the most suitable transformations that linalool can undergo at acidic conditions (low pH) and high temperatures, such as those found in the extraction process [85,143,144]. The variation in the chemical composition of EO and its dependency on the parameter conditions, such as those in the pretreatment stage, have not been studied widely, and they are not even deeply discussed. Therefore, it is expected that a deep insight into the relationship between the chemical composition and extraction parameters will be assessed in future studies when NADES are employed in the extraction of EO.

As described previously, the use of NADES significantly affects EO extraction, specifically diluting the cell wall components and releasing the cellular compounds. However, NADES viscosity is one of the main factors that can limit the extraction of EO, and even when water is added, poor permeability is reflected in low yields and the extraction of fewer compounds [145]. Therefore, the use of enzymes during pretreatment could be a helpful strategy for enhancing the extraction process [146]. In this sense, during the extraction of EO from *Mentha haplocalyx* Briq. leaves, biological enzymes (cellulase, pectinase, hemicellulase, papain, and xylanase) were used during the pretreatment stage (enzymolysis) combined with NADES, and parameters such as enzyme type, NADES type, water content, enzyme concentration, enzymolysis temperature, time, and pH were optimized. The results showed that the ideal extraction solvent was ChCl/Glu with an 80% water content. Additionally, a significant increase in EO yield was observed for the five evaluated enzymes, with the highest yield observed in the mixed enzyme assay (cellulase and pectinase), reaching a yield of 1.75% compared with 1.50% obtained without enzyme pretreatment. Regarding the chemical composition, menthone content in EO was lower in all the pretreated assays with enzymes than in pretreatment without. At the same time, menthol increased significantly, and D-limonene was only found in the mixed enzyme assays, highlighting the potential use of more than one enzyme during the pretreatment stage. Enzyme concentration significantly increased EO yield up to 2.0% when other important factors such as pH, temperature, and pretreatment time (enzymolysis) were optimized, and the menthol concentration showed its highest value, as did caryophyllene, germacrene D, and menthyl acetate [96].

Enzymolysis time, pH, and temperature influence the EO yield and chemical composition. This was observed when the enzymolysis temperature of 50 °C led to the highest extraction yield and decreased with higher temperatures (1.81%). Furthermore, the chemical composition varied according to temperature, and there was no observable pattern between the composition of the main compounds in the EO. Significant differences were observed when the pH value varied during enzymolysis, reaching the highest EO yield at pH 4.0. This factor also influenced the chemical composition, where increasing the pH increased the proportion of menthol and germacrene D and decreased menthone and isomenthol. In contrast, piperitone and its oxide were significantly reduced at lower pH values. Lastly, EO yield increased with enzymolysis time until 4.0 h, reaching its maximum value at this time, decreasing considerably in the following hours, and increasing at 10.0 h, which is attributed to the time required for the complete hydrolysis of the plant material under the previously optimized conditions (temperature, pH, enzyme type, and concentration) [147]. The chemical composition varied similarly, where piperitone, caryophyllene, and menthol content reached their highest values at 4.0 h while menthone, pulegone, d-limonene, and piperitone oxide decreased first and increased close to 10.0 h of enzymolysis time [96]. These are essential aspects of enzyme functionality that impact its activity because enzymes are biological molecules that can be denatured if exposed to rough conditions such as high temperatures, non-working pH values, and long exposure times [148].

Lastly, enzymolysis combined with NADES during pretreatment resulted in a synergistic and positive effect on EO extraction of *Mentha haplocalyx*, yielding the highest amount (2.56%) compared to water-based enzymolysis (1.84%) and NADES-based non-enzymatic pretreatments (2.19%) [96]. According to the authors, this improvement could be attributed to the ability of NADES to stabilize and support the enzyme, maintaining its activity for extended periods, and enhancing the extraction process. Additionally, the solubilizing capacity of NADES facilitates the permeation of cell walls and their components, thereby promoting the release of EO components into the medium [149,150]. The chemical composition of *M. haplocalyx* EO was similar across different methods, with menthol, menthone, iso-menthone, and piperitone as the main components, although their proportions varied.

As the three main aspects related to the extraction of essential oils with NADES have been covered (Figure 4), it is also crucial to examine the key characteristics and applications of the oils obtained.

## 5. Analytical Methods Used for the Characterization of Essential Oils Components and Some of Their Biological Activities

During the final step of the extraction process, the essential oil is separated and weighed to determine the yield. Its chemical composition is then analyzed using gas chromatography coupled with mass spectrometry (GC-MS) and/or flame ionization detection (GC-FID). Gas chromatographic techniques are particularly suitable for EO analysis because, as discussed earlier, the chemical composition can vary significantly depending on the extraction conditions at each step. Moreover, these techniques effectively handle the volatile nature, low molecular weight (approximately < 300 amu), and diverse composition of EOs [151], as demonstrated in many studies included in this work (see Table 1). Additionally, through the different studies, it was observed that this technique can be employed to identify compounds that were extracted exclusively using NADES. These techniques also monitor changes in the proportion of specific compounds of interest and the number and proportion of oxygenated hydrocarbons in the total composition when NADES are used.

In addition to conventional chromatographic methods, such as GC-MS, headspace (HS) and solid-phase microextraction (SPME) analysis offer valuable insights into the volatile compounds in essential oils extracted using NADES. SPME is particularly effective for detecting trace amounts of volatile compounds, which are crucial for the aroma and overall profile of the EO. The NADES extraction and dilution enhanced the sensitivity and efficiency of the analysis. For example, ChCl/Glu (1:1) has been demonstrated to be a prominent solvent for the extraction and analysis of volatile compounds in sweet leaves of *Ipomoea cairica* (L.), where a total of 77 volatile compounds were identified, of which 43% were terpenoid, 35% aromatic, and 22% aliphatic compounds, with β-elemene, β-caryophyllene, and α-humulene as the main compounds [109]. In the analysis of tobacco (*Nicotiana tabacum* L.) volatiles, the synergy between microwave radiation and NADES extraction capacity was used to identify a higher number of compounds by SPME (65) compared to water-based extraction (47) and headspace (34). Additionally, the method reduced extraction time, showed high efficiency, and could even differentiate tobacco samples from different geographical regions [152]. Similarly, NADES were used during the analysis of volatile compounds of peppermint leaves (*Mentha piperita* L.) under HS-SPME conditions after an ultrasound-assisted extraction, where ChCl/Glu (5:2) showed the highest efficiency in the extraction of compounds such as menthol, menthone, menthyl acetate, eucalyptol, and pulegone, and was also used for the determination of those compounds in samples from five different countries [116]. In another study, a headspace solvent microextraction (HS-SME) approach was used to analyze the contents of linalool, α-terpineol, and terpinyl acetate with the combination of ChCl/EG in different molar ratios (1:2–5) [86]. The results indicated that, compared to other methods (ultrasound-assisted methanol-based extraction and heating methanol extraction), the NADES-based approach contributed to a more remarkable recovery of linalool and α-terpineol and reduced extraction times and costs [86]. In summary, NADES provides an alternative to conventional solvents for the analysis of volatile compounds. They enhance the sensitivity of the methods by identifying compounds that might be degraded during traditional EO extraction methods and offer a more efficient, cost-effective, and environmentally friendly option for EO component analysis.

Once the detailed chemical profile of EO has been established through chromatographic analysis, it is essential to explore its biological activities and other notable features. Evaluating the biological properties of EOs provides valuable insights into their potential applications. For instance, antioxidant activity has been assessed by different methods, such as the scavenging capacity of DPPH and ABTS radicals, ferric reducing antioxidant power (FRAP), and the content of phenolic compounds in EOs from manuka (*Leptospermum scoparium*) leaves [103], lemongrass species (*Cymbopogon citratus*, *Cymbopogon flexuosus*, and *Elyonurus muticus*) leaves [97,104], and rosemary (*Rosmarinus officinalis* L.) leaves [114], where the use of NADES increased the antioxidant activity compared with water-based extraction. The antibacterial effect on gram-positive bacteria of EOs extracted with NADES showed significant differences from EOs obtained without NADES, attributed mainly to the components extracted and their proportions, as has been observed in the EO from perilla (*Perillae folium*) leaf [95] and Mexican lippia (*Aloysia citriodora*) leaves [107].

The anti-α-glucosidase activity of *Zanthoxylum bungeanum* EO suggests that it could be used as a hypoglycemic drug since, according to molecular docking, its main components (terpinene-4-ol, β-pinene, terpinyl acetate, α-terpineol, linalool, and limonene) combine with α-glucosidase and inhibit the decomposition of starch polysaccharides [153]. The insecticidal activity of *Nardostachys jatamansi* (D.Don) DC EO varied according to the composition of NADES, with ChCl/LA NADES at 1:1 and 1:2 proportions showing toxicity against *A. craccivora* after 72 and 96 h of topical contact, respectively, and was comparable to that of azadirachtin (positive control). Similarly, EOs obtained with ChCl/MA (2:1) and glycerol and lactic acid (1:1) showed the highest toxicity against *P. lilacinus* and were comparable to the positive control [113]. Substances that exhibit anti-acetylcholinesterase (AChE) activity are potential ingredients in insecticidal formulations [154]. In addition to the enhanced inhibitory effect of *Nardostachys jatamansi* EO, *Mentha haplocalyx* Briq. leaves and *Ageratina adenophora* flower EOs showed superior anti-acetylcholinesterase activity when using enzyme NADES-based pretreatment and NADES-based extraction compared to their analogous NADES-free extraction methods [96,99]. Lastly, it is important to mention that, although NADES offer advantages in the extraction of bioactive compounds, they also can interfere during the assessment of their biological activities through in vitro assays, and therefore further extractions are necessary; for example, during the extraction of terpenoids and polyphenols from *Citrus reticulate* leaves, a solid-phase extraction (SPE) step was necessary after the extraction under UA-NADES conditions in order to remove NADES from the extracts since they interfered during the quantification of phenolic compounds and the AChE assay [155].

In addition to the interferences regarding in vitro assays produced by the presence of NADES in the extracts, the sole presence of NADES components can hardly compromise its use in pharmaceuticals, food, and cosmetics, especially since those secondary metabolites will have a larger molar ratio than the usual in the plant matrix [34]. Furthermore, the toxicity of NADES in humans is still under extensive examination. Given that NADES formulations can be tailored by varying their components, proportions, and synthesis methods to create unique solvents for specific extraction conditions, it becomes crucial that toxicity assays are carried out in every study. The variability in NADES formulations means that each new solvent may present different toxicological risks, making them unsuitable as ready-to-use delivery systems for bioactive compounds intended for human use. Consequently, bio-refinery processes become necessary as an additional step to ensure the safe extraction of bioactive compounds with NADES. [45,156,157,158,159,160]. For example, in the work of Domínguez-Rodríguez et al. [155] and Cokdinleyen et al. [161], a step of the bio-refinery approach previous to analysis is solid-phase extraction (SPE) of the bioactive compounds extracted with NADES, this due to the interferences that NADES may cause in the assays and also because one of the objectives is the recovery of the highest amount of bioactive compounds. Therefore, NADES extraction not only serves as an ultimate tool during the extraction but also as an important step before the recovery and isolation of bioactive compounds, taking advantage of their extractive benefits for improving the overall extraction process, reducing the potential toxicity of NADES components, and also offering an almost pure extract [162].

Although all the studies included in the work focused almost exclusively on extraction, characterization, optimization, and biological activity assessment of EOs extracted with NADES, the stability of EO components was one of the features almost unexplored in all of the studies collected. It is expected that they also contribute to preserving EOs since NADES are biodegradable and chemically inert and have been shown to stabilize volatile compounds and their biological activities [77], highlighting their use as green solvents.

## 6. Future Perspectives and Conclusions

The use of NADES in extracting EOs represents an advantageous alternative to conventional and advanced methods in terms of efficiency, cost-effectiveness, biodegradability, and even operational simplicity. Additionally, their tailoring capacity makes them versatile tools during extraction, offers high variability of components, and is useful in the extraction of EOs or their specific components. NADES have been widely studied with different approaches, being their extractive capacities the most valuable feature that promotes their use. Regarding the extraction of EOs, the present review collects several studies about the main factors involved when using NADES in the extraction of EOs, the techniques employed, and the different approaches used to improve the overall extraction. The tailoring of NADES (e.g., components, molar ratio, and water content), techniques such as microwave-assisted and ultrasound-assisted, and their combination with conventional procedures like hydro-distillation for the separation of EOs and the application of pretreatment to the plant material for improving the overall extraction procedure are the main approaches employed in the extraction of EOs with NADES. ChCl is the main HBA used in NADES composition, and it is usually combined with polyols and organic acids that lead to the production of hydrophilic solvents, suitable for the extraction of terpenoids, valuable compounds due to their biological properties. However, only laboratory-scale extractive approaches have been conducted, where the optimization of those parameters seems to be the critical goal that needs to be accomplished in the field. The remaining substantial factors, such as the purity of the extracted EOs, environmental impact, cost and energy efficiency during the extraction, NADES reusability, and prospects regarding the scale-up of the process and their operational implications, were absent in all of the studies reviewed.

Due to the above, although the use of NADES and advanced techniques in the extraction of EOs has shown promising results in laboratory environments and offers sustainable benefits compared with conventional methods, there is still the need for further development and research in order to refine extraction of EOs to make them more sustainable and efficient before being applied in the industry. Factors such as NADES viscosity, the tailoring of NADES components and properties, the cost and environmental impact related to the amount of EO isolated, the energy and mass transfer during the extraction process, the separation from the extraction medium, the optimization of the operational parameters and even the reusability and stability of EOs components in NADES represent exciting hotspots for future research that need to be further developed and widely studied since EOs are of particular interest for the pharmaceutical and cosmetic industry and these will be critical factors for a proper scaling-up process of EOs extraction. Therefore, questions such as if the use of NADES in the extraction of EOs is indicated from an economical and sustainable perspective and if the advantages offered by the use of NADES will justify the additional investment that possibly will require the scaling-up process must be answered first in order to make a proper transference to the industrial level. Lastly, although NADES toxicity is under study, the exploration for other HBAs and HBDs that can be used in products designed for humans is also another research opportunity that may take advantage of the biological properties of EOs components in the design of delivery systems or ready-to-use formulations, adding a new layer of value to the EOs obtained with NADES.

## Figures and Tables

**Figure 1 molecules-30-00284-f001:**
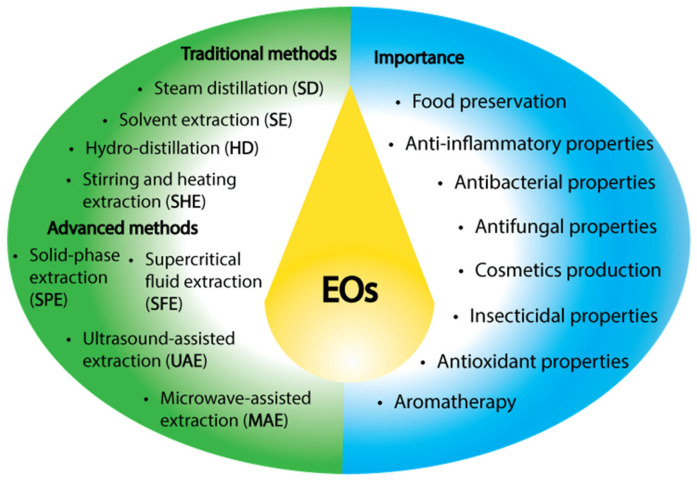
Main properties and extraction methods for EOs.

**Figure 3 molecules-30-00284-f003:**
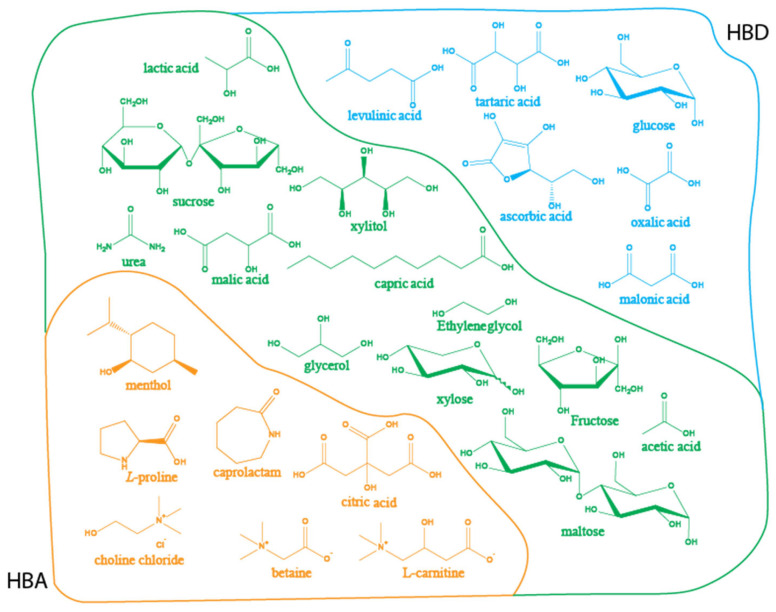
Main HBAs (in yellow zone) and HBDs (in blue zone) used in NADES during the extraction of EOs. Compounds found in the green zone were used indistinctively as HBA and HBD.

**Figure 4 molecules-30-00284-f004:**
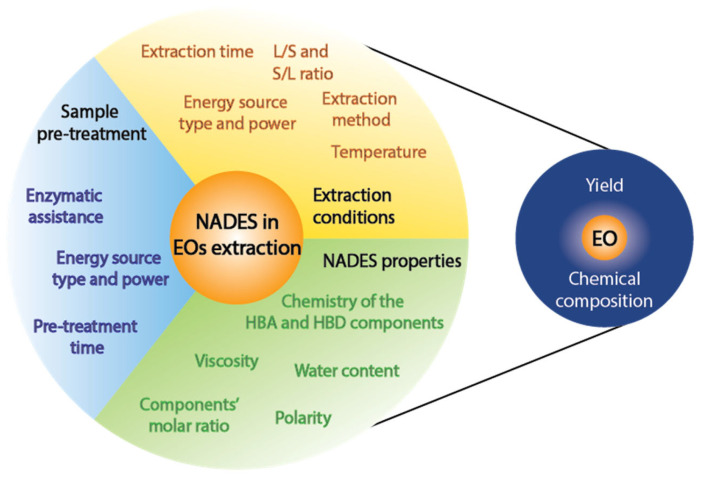
Key parameters in EO extraction using NADES.

## Data Availability

Not applicable.

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
