# Peer review of "Exploring Natural Deep Eutectic Solvents (NADES) for Enhanced Essential Oil Extraction: Current Insights and Applications"

_molecules, 2025, doi:10.3390/molecules30020284_

Round 1
Reviewer 1 Report
Comments and Suggestions for Authors
This is a sounding theme, and the use of NADES for extraction of essential oils and their stabilization has long been studied. This review is appropriate in collecting the body of work relative to EOs extraction with NADES, but does not comment on any of the works, and the future perspectives and conclusions section could have been more developed, presenting more solutions for the future of NADES and EOs.
I only have some minor comments:
- line 34, the term "aromatic" could be changed by "fragrant"
-line 145, NADES similarity to some intracellular media, is of course one of the characteristics that make them able to solubilize components such as the ones present in EOs, but this is not the only reason. I would comment on their interactions with the EOs components, as well as the fact that NADES can have a tailor made compositions, and be more specific for the extraction of a compound of interest. Also, instead ou "substance" change to "mixtures"
- Maybe this is an option of the journal or because it is the first version, but et al. should be in Italic, throughout the text
-Starting in line 169, in all of the examples of extraction of EOs you mention, the used NADES have a more polar nature, and have quite low extraction yields. Do you think that using ChCl and sugar based NADES is the best option to extract EOs rich in terpenoids? Thy extract also some polyphenols and in that way I understand. But, is there not a way to find a more specific NADES to target extraction of the EO main components? What about polarity compatibility? A comment on this could be added, like the one started in line 242
- Following the previous comment, ChCl, oxalic acid, malonic acid (for example) are often not allowed in the final formulation of cosmetic, food or pharmaceutical products. What would you comment on this? Are there alternatives?
- I am missing in some of the extraction examples given and cited, how much does actually NADES extraction increases the yield, when compared to traditional extraction. I have only seen this comparison in microwave assisted extraction.
- Figure 4, in the NADES properties, maybe it would be good to add polarity.
- The separation of the EO from the extraction Solvent (in this case NADES) is not sufficiently discussed. This step is crucial and ideas on how this separation process occurs, would be good to see in this manuscript (mainly related to solvent sustainability and separation process cost and time).
Author Response
Dear Reviewer 1
Thank you very much for taking the time to review this manuscript. Please find the corrections highlighted in the re-submitted file.
- This is a sounding theme, and the use of NADES for extraction of essential oils and their stabilization has long been studied. This review is appropriate in collecting the body of work relative to EOs extraction with NADES, but does not comment on any of the works, and the future perspectives and conclusions section could have been more developed, presenting more solutions for the future of NADES and EOs.
Response: The future perspectives and conclusion section was renovated in order to express the perspective of the authors concerning the information collected and consigned in the review and the future of the research of NADES in the extraction of EOs, pointing out the future research opportunities that arise from the research gaps found during the review process.
I only have some minor comments:
- line 34, the term "aromatic" could be changed by "fragrant"
Response: The term has been changed and highlighted.
- line 145, NADES similarity to some intracellular media, is of course one of the characteristics that make them able to solubilize components such as the ones present in EOs, but this is not the only reason. I would comment on their interactions with the EOs components, as well as the fact that NADES can have a tailor made compositions, and be more specific for the extraction of a compound of interest. Also, instead ou "substance" change to "mixtures"
Response: The term “substance” has been replaced with “mixtures”. Additionally, NADES interactions with EOs components and some examples of the tailoring of these solvents in the extraction of specific EOs components were included.
- Maybe this is an option of the journal or because it is the first version, but et al. should be in Italic, throughout the text
Response: The style of the abbreviation for other authors has been changed to Italic.
- Starting in line 169, in all of the examples of extraction of EOs you mention, the used NADES have a more polar nature, and have quite low extraction yields. Do you think that using ChCl and sugar based NADES is the best option to extract EOs rich in terpenoids? Thy extract also some polyphenols and in that way I understand. But, is there not a way to find a more specific NADES to target extraction of the EO main components? What about polarity compatibility? A comment on this could be added, like the one started in line 242
Response: The authors agree with this particular observation and we consider that regarding exclusively the components of NADES, the chemical composition impacts greatly the extracting medium and indicates which compounds will be extracted. Due to this, the section where the ChCl-based NADES with sugars as HBDs was extended including examples of the tailoring of NADES for the extraction of target compounds (both, hydrophilic and hydrophobic compounds) and its relation with the polarity and viscosity was also included.
- Following the previous comment, ChCl, oxalic acid, malonic acid (for example) are often not allowed in the final formulation of cosmetic, food or pharmaceutical products. What would you comment on this? Are there alternatives?
Response: In line 837 a paragraph was included regarding the use of NADES-extracted compounds in human products such as food or pharmaceuticals.
- I am missing in some of the extraction examples given and cited, how much does actually NADES extraction increases the yield, when compared to traditional extraction. I have only seen this comparison in microwave assisted extraction.
Response: In the three sub-sections of section four, specific details were added on how NADES presence or absence, the use of advanced extraction techniques with conventional methods, and the inclusion of a pre-treatment stage during the extraction impacted the EO yield.
- Figure 4, in the NADES properties, maybe it would be good to add polarity.
Response: “NADES’ polarity” was added to the “NADES properties” section. Thanks for the recommendation.
- The separation of the EO from the extraction Solvent (in this case NADES) is not sufficiently discussed. This step is crucial and ideas on how this separation process occurs, would be good to see in this manuscript (mainly related to solvent sustainability and separation process cost and time).
Response: As authors, we agree and appreciate this observation regarding our work. That’s why in sub-section number two from section four was added discussion regarding the separation process after the extraction with NADES and the economic and sustainable perspectives related to the extraction with these solvents.
Reviewer 2 Report
Comments and Suggestions for Authors
This review provides a comprehensive overview of the use of NADES (Natural Deep Eutectic Solvents) in essential oil (EO) extraction, detailing methods, optimization strategies, and potential applications. While the topic remains relevant within the context of green chemistry, the authors present NADES as a relatively new innovation (Line 71). This is a misleading characterization, as research on NADES has been ongoing for years. Consequently, this topic can no longer be considered novel, even though significant research is still ongoing in this field. Apart from this, there are several areas in the manuscript that require improvement, along with some mistakes that need to be addressed.
Firstly, the manuscript should be carefully reviewed for empty spaces, such as the one noted on line 69, and unnecessary empty lines, such as line 116, which should be removed to ensure consistency and proper formatting.
Additionally, captions and labels for figures and tables need to be justified to maintain a uniform and professional appearance throughout the document. Furthermore, the reference style should be thoroughly checked.
In addition, the description of the procedures used in Table 1 for optimizing NADES formulations lacks critical details. While the authors summarize various NADES applications, specific experimental conditions such as the energy source (e.g., ultrasound or microwave), power settings, durations, and temperatures are inconsistently reported or omitted. Furthermore, in some cases, exact solvent-to-solid ratios (L/S or S/L) and water content in NADES formulations are not provided, despite their significant influence on extraction efficiency and yield.
Moreover, the review references studies employing response surface methodology (RSM) for optimization but does not provide adequate explanations of the factors selected for optimization, the specific statistical models used, or the resulting optimal conditions.
In the discussion of microwave-assisted hydro-distillation (MAHD) and NADES-assisted techniques, the authors claim these methods achieve significantly higher yields and better retention of bioactive compounds compared to conventional hydro-distillation. However, these claims are not consistently supported by experimental evidence or citations, weakening the validity of the conclusions drawn.
Figures 2 and 3 require significant revision. The molecular structures depicted are difficult to interpret due to the dark background and low-contrast colors. Using a white background and employing high-contrast colors for the structures would substantially improve their visibility and comprehension for readers.
The manuscript also contains several grammatical errors and unclear expressions that reduce its overall readability. For example, in line 167, the sentence "switching from different NADES components, their molar amounts in the formulation, and the water content modulates and extensively influences the composition" is awkwardly phrased. A clearer revision could be: "Modifying NADES components, their molar ratios, and water content significantly influences composition and extraction efficiency." Similarly, in line 297, the statement that NADES "can transform this electromagnetic energy into thermal energy, thereby disrupting the cell walls" could be streamlined for clarity and precision.
Moreover, the review repeatedly emphasizes the use of ChCl-based NADES without providing a robust discussion of their comparative performance against other NADES formulations or traditional solvents. A critical analysis of alternative formulations and their relative effectiveness is necessary to provide a balanced perspective.
Finally, while the authors explore the potential of NADES to improve EO yield and bioactive compound retention, they do not sufficiently address the scalability and economic feasibility of these methods. For example, the discussion of NADES viscosity and its impact on large-scale extraction processes is superficial. Although water content is mentioned as a means of reducing viscosity, the authors do not explore the practical implications of this adjustment for industrial applications. A more detailed examination of the challenges and opportunities associated with scaling up NADES-based extraction methods would enhance the manuscript’s practical relevance.
Author Response
Dear Reviewer 2
Thank you very much for taking the time to review this manuscript. Please find the corrections highlighted in the re-submitted file.
- This review provides a comprehensive overview of the use of NADES (Natural Deep Eutectic Solvents) in essential oil (EO) extraction, detailing methods, optimization strategies, and potential applications. While the topic remains relevant within the context of green chemistry, the authors present NADES as a relatively new innovation (Line 71). This is a misleading characterization, as research on NADES has been ongoing for years. Consequently, this topic can no longer be considered novel, even though significant research is still ongoing in this field. Apart from this, there are several areas in the manuscript that require improvement, along with some mistakes that need to be addressed.
Response: We greatly appreciate this correction and consider that it has been more than ten years since natural DES was proposed as an extracting media. Due to this, the term “novel” in line 71 has been changed to “greener,” and a small commentary regarding the scientific production related to NADES use was added in the next paragraph.
- Firstly, the manuscript should be carefully reviewed for empty spaces, such as the one noted on line 69, and unnecessary empty lines, such as line 116, which should be removed to ensure consistency and proper formatting.
Response: We appreciate the observation, empty spaces and lines have been removed from the manuscript.
- Additionally, captions and labels for figures and tables need to be justified to maintain a uniform and professional appearance throughout the document. Furthermore, the reference style should be thoroughly checked.
Response: We appreciate the observation, captions and labels for the figures and the table have been justified. Additionally, it has been checked that the style of the abbreviation for other authors (et al.) is in Ithalic.
- In addition, the description of the procedures used in Table 1 for optimizing NADES formulations lacks critical details. While the authors summarize various NADES applications, specific experimental conditions such as the energy source (e.g., ultrasound or microwave), power settings, durations, and temperatures are inconsistently reported or omitted. Furthermore, in some cases, exact solvent-to-solid ratios (L/S or S/L) and water content in NADES formulations are not provided, despite their significant influence on extraction efficiency and yield.
Response: We appreciate the observation and specific experimental conditions were added to the table.
- Moreover, the review references studies employing response surface methodology (RSM) for optimization but does not provide adequate explanations of the factors selected for optimization, the specific statistical models used, or the resulting optimal conditions.
Response: We appreciate the observation and the section regarding the use of RSM in the optimization approaches of the extraction of EOs with NADES was extended.
- In the discussion of microwave-assisted hydro-distillation (MAHD) and NADES-assisted techniques, the authors claim these methods achieve significantly higher yields and better retention of bioactive compounds compared to conventional hydro-distillation. However, these claims are not consistently supported by experimental evidence or citations, weakening the validity of the conclusions drawn.
Response: As authors, we sincerely appreciate this observation and it is correct, the EO yield obtained with advanced techniques such as water-based MAHD is lower than conventional HD. However, although carbon footprint and energy cost of extraction are not included in the study and therefore it is not possible to show evidence regarding the superiority of advanced techniques such as MAHD and UAHD over conventional HD with this example, in the development of the same sub-section, it was included the energy cost and the environmental impact of extracting EOs with the same techniques where the superiority of the advanced techniques was observed in terms of the quantity of EO extracted per energy consumed (line 633), supporting the conclusion drawn in the first version of the manuscript.
Lastly, it is important to mention that the NADES-based MAHD showed the highest yield compared with other techniques and solvents, highlighting their use in the extraction of EOs.
- Figures 2 and 3 require significant revision. The molecular structures depicted are difficult to interpret due to the dark background and low-contrast colors. Using a white background and employing high-contrast colors for the structures would substantially improve their visibility and comprehension for readers.
Response: Both figures were modified according to the suggestion. Thank you for your observation.
- The manuscript also contains several grammatical errors and unclear expressions that reduce its overall readability. For example, in line 167, the sentence "switching from different NADES components, their molar amounts in the formulation, and the water content modulates and extensively influences the composition" is awkwardly phrased. A clearer revision could be: "Modifying NADES components, their molar ratios, and water content significantly influences composition and extraction efficiency." Similarly, in line 297, the statement that NADES "can transform this electromagnetic energy into thermal energy, thereby disrupting the cell walls" could be streamlined for clarity and precision.
Response: We greatly appreciate the reviewer’s valuable observations regarding the grammatical errors in the manuscript. In addition to addressing the specific issues highlighted, we have thoroughly revised the entire review to improve its clarity and readability.
- Moreover, the review repeatedly emphasizes the use of ChCl-based NADES without providing a robust discussion of their comparative performance against other NADES formulations or traditional solvents. A critical analysis of alternative formulations and their relative effectiveness is necessary to provide a balanced perspective.
Response:
Thank you for your insightful comment. We appreciate the importance of comparing ChCl-based NADES with other formulations and traditional solvents. However, the studies included in this review primarily focus on the overall performance of the NADES as solvent systems, rather than isolating and analyzing the individual components. In many cases, the studies either do not provide detailed information on the individual components of the NADES or make combinations that are not directly comparable, which complicates a meaningful analysis of alternative formulations.
Furthermore, while there are some studies that explore the use of alternative HBAs to ChCl, these studies are limited and, in most cases, do not offer sufficient or consistent data to allow for a robust comparison of extractive performance across different NADES systems. As a result, it is difficult to conduct a critical analysis of non-ChCl-based NADES formulations within the scope of this review. Nevertheless, we have included comparisons between different ChCl-based NADES formulations and traditional solvents, which we believe provide valuable insights into their relative performance. We hope this approach effectively addresses the reviewer’s comment while maintaining the focus of the review.
- Finally, while the authors explore the potential of NADES to improve EO yield and bioactive compound retention, they do not sufficiently address the scalability and economic feasibility of these methods. For example, the discussion of NADES viscosity and its impact on large-scale extraction processes is superficial. Although water content is mentioned as a means of reducing viscosity, the authors do not explore the practical implications of this adjustment for industrial applications. A more detailed examination of the challenges and opportunities associated with scaling up NADES-based extraction methods would enhance the manuscript’s practical relevance.
Response: We appreciate the observation and agree that it’s necessary to address the scalability and economical features of the extraction of EOs with NADES. Due to this, the separation of EOs from the extracting media and the economic and environmental impact of the extraction process were discussed in sub-section two from section four. Additionally, the possible implications and perspectives regarding future research and the possible challenges of a transition to an industrial level were addressed in the perspectives and conclusions section.
Round 2
Reviewer 2 Report
Comments and Suggestions for Authors
The authors have addressed all the points previously mentioned in a thorough manner, and I believe the manuscript is now prepared for publication. However, I recommend a minor revision: please alter the color of the compounds displayed in Figure 3, as yellow may be challenging for readers to discern. Additionally, efforts should be made to eliminate large empty spaces, such as the space between line 327 and Table 1. Please also remove empty lines, eg. lines 469 and 460, as well as 1075-1076. From Figure 4, the term 'NADES' have ' that is not required. Also, I a careful review and necessary corrections of the Authors' Contributions section at the end of the paper, as well as the Informed Consent Statement and Data Availability Statement is needed. Additionally, please remove any empty spaces between the references.
Author Response
Dear Reviewers,
We sincerely appreciate your valuable comments and feedback. In response to your suggestions, we have made the following revisions in the final version of the manuscript:
- The color of the compounds in Figure 3, previously displayed in yellow, has been changed to a darker shade for improved visibility.
- A portion of Section 4 has been repositioned before Table 1 to eliminate the large empty space and to ensure better flow of content.
- Empty lines (lines 459, 460, 747, 1075, and 1076) as well as unnecessary spaces between paragraphs have been removed.
- In Figure 4, the term "NADES" was removed from the green zone labeled "NADES properties," as repeating "NADES" was redundant given that the zone's name already refers to these solvents.
- The extra spaces between references have been eliminated to ensure proper formatting.
- Sections such as the supplementary materials, authors' contributions, Informed Consent Statement, Data Availability Statement, and Conflicts of Interest have been carefully reviewed and updated.
We would like to express our sincere gratitude for your thoughtful and constructive review, especially considering the time of year. Your efforts in improving the quality of our work are greatly appreciated.
Kind regards,